# Prospective Learning: Memory-Efficient MLP Training via Brain-Inspired Direct Optimization

## Abstract

Multi-layer perceptron (MLP) training via backpropagation faces fundamental memory limitations that constrain deployment in resource-constrained environments such as edge devices. We introduce Prospective Learning, a novel training paradigm inspired by biological prospective configuration mechanisms that replaces gradient-based optimization with direct algebraic weight computation. By transforming weight updates into regularized least-squares optimization problems that can be solved analytically layer by layer, it eliminates the need for gradient storage and intermediate activation caching, significantly reducing resource consumption. Meanwhile, it integrates brain-inspired sparse connectivity initialization and adaptive metaplasticity mechanisms, which support the framework from the aspects of infrastructure initialization and dynamic learning adjustment, respectively. Experiments on the MNIST, CIFAR-10, and CIFAR-100 datasets show that Prospective Learning achieves competitive accuracy, reduces memory usage by up to 55% compared with traditional backpropagation, and consistently outperforms existing backpropagation alternatives in memory efficiency. This memory-computation trade-off is favorable for edge scenarios where memory constraints dominate. For example, it achieves 95.44% accuracy on MNIST using only 38.77MB of memory on edge devices, providing a viable solution for efficient MLP training on memory-constrained edge devices. Our main code has been anonymously uploaded to `https://anonymous.4open.science/r/Prospective-Learning` without any author information.

## 1 Introduction

Deep neural networks trained via backpropagation (BP) have achieved remarkable success across diverse domains, from computer vision to natural language processing (Kalluri et al., 2025; Luo et al., 2025; Liang et al., 2025; Consens et al., 2025). The backpropagation algorithm operates through a two-phase process: forward propagation computes predictions and loss functions, while backward propagation calculates gradients to update network weights based on error signals (Rumelhart et al., 1986). However, this gradient-based learning paradigm faces fundamental challenges that limit its broader applicability. Backpropagation requires storing intermediate activations for gradient computation, leading to substantial memory overhead that scales linearly with network depth. These limitations become particularly problematic in resource-constrained environments such as memory-constrained edge devices, where resource-limited hardware struggles to accommodate the computational demands of gradient-based training while biological computing paradigms demand fundamentally different learning approaches (Lillicrap et al., 2020; Shuvo et al., 2022; Wang et al., 2025).

Alternative learning paradigms that reduce reliance on traditional backpropagation can be broadly categorized into several approaches: perturbation-based method (Fernandez et al., 2024) that introduces random noise to estimate gradients through finite differences, decomposition-based techniques such as alternating direction method of multipliers (ADMM) (Wang et al., 2019) that break training into sequentially solvable subproblems, and forward-only algorithms exemplified by the Forward-Forward (FF) method (Hinton, 2022) that replaces backward passes with contrastive forward passes using positive and negative data samples. More recently, NoProp emerged as a diffusion-inspired

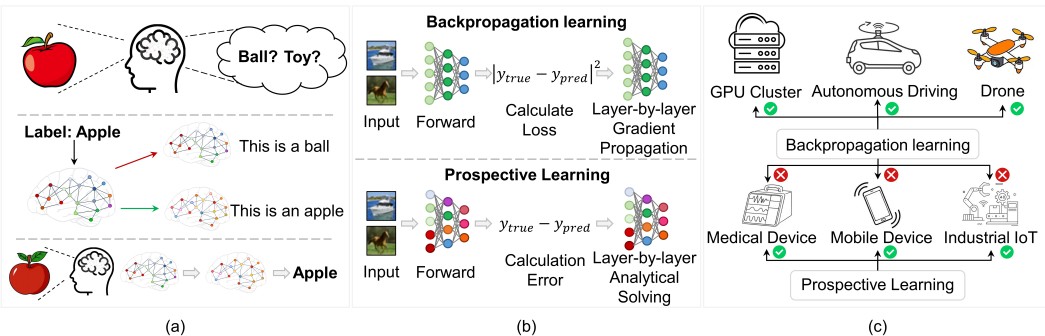

Figure 1: **Prospective Learning Overview.** (a) Human learning first infers target brain activity patterns then updates neural connections to achieve these patterns. (b) Traditional algorithms use layer-by-layer gradient backpropagation while prospective learning solves weights analytically without weight gradient computation. (c) Backpropagation-based methods fail in resource-constrained environments whereas prospective learning enables training in memory-limited edge scenarios.

paradigm that reframes neural network training as a layer-wise denoising process (Li et al., 2025), where each layer learns to reconstruct clean activations from noisy inputs through iterative refinement steps. However, these approaches face significant limitations that have hindered their practical adoption. Perturbation-based methods suffer from poor scalability and training instability, while decomposition techniques often require problem-specific formulations that limit their generalizability. Although NoProp demonstrates comparable performance on vision benchmarks, experimental analysis reveals that it still relies on gradient computations during training phases and maintains substantial memory requirements for storing denoising targets and intermediate states. Consequently, existing backpropagation alternatives remain inadequate for replacing gradient-based training in resource-constrained environments, where memory limitations represent the primary bottleneck that determines deployment feasibility.

Recent advances in brain-inspired algorithms have demonstrated remarkable success by incorporating computational principles derived from neuroscientific discoveries. Research into biological neural mechanisms has transformed artificial intelligence algorithm design, with insights from synaptic plasticity enabling more adaptive learning systems (Kudithipudi et al., 2022), neural oscillation patterns improving temporal processing capabilities (Miikkulainen, 2025), and cortical organization principles enhancing network architectures for tasks ranging from pattern recognition to sensorimotor control (Pulvermüller et al., 2021).

Particularly promising is the application of brain-inspired mechanisms to alternative learning approaches, as neuroscientific investigations have revealed the existence of prospective configuration mechanisms in biological neural networks (Song et al., 2024), where learning occurs through a two-stage process: neurons first infer target activity patterns based on desired outcomes, followed by synaptic weight adjustments that solve for these target configurations. This prospective learning paradigm offers significant advantages over traditional error-driven approaches, including reduced computational interference between learning signals, more direct credit assignment to individual synapses, and enhanced memory efficiency through reduced gradient computation and storage requirements, presenting tremendous potential for addressing the resource limitations inherent in conventional neural network training.

Inspired by biological prospective configuration mechanisms, we present Prospective Learning, a novel learning framework that replaces gradient-based weight updates with direct algebraic optimization while preserving the essential error propagation structure of neural network training. Our approach comprises three key technical contributions: (1) a prospective configuration algorithm that employs least-squares optimization to directly compute optimal weight updates without weight gradient computation or storage; (2) Sparse Connectivity Initialization, which reduces network parameters while maintaining the network's representational capacity; and (3) Adaptive Metaplasticity Mechanism, which achieves dynamic balance optimization for prospective learning. These three components work in synergy to reduce memory usage while enabling efficient learning. Figure 1

illustrates this fundamental paradigm shift from gradient-based to prospective learning mechanisms and their respective deployment scenarios.

Extensive experimental validation demonstrates that PL achieves comparable accuracy while dramatically reducing memory consumption compared to traditional gradient-based methods. Across benchmark datasets including MNIST, CIFAR-10, and CIFAR-100, our framework maintains competitive performance relative to backpropagation baselines while reducing memory usage by up to 55% through elimination of gradient storage and intermediate activation caching. Our approach consistently outperforms other backpropagation alternatives in memory efficiency, with strong results on edge devices where we achieve 95.44% accuracy on MNIST using only 38.77MB memory. These results demonstrate that algebraic optimization combined with brain-inspired connectivity patterns provides a compelling solution for memory-constrained edge deployment, where the per-layer computational cost represents a favorable trade-off given that memory limitations dominate deployment feasibility.

## 2 RELATED WORK

### 2.1 BACKPROPAGATION ALTERNATIVES AND STRUCTURED LEARNING

Alternative learning paradigms represent a paradigm shift away from traditional backpropagation-based training, aimed at addressing fundamental limitations including memory overhead, computational complexity, and biological implausibility of gradient computation (Flaxman et al., 2004; Duchi et al., 2015; Bolager et al., 2024). These approaches broadly encompass several methodological directions, each offering distinct advantages and limitations. Specialized architectural optimization approaches (Van Thieu et al., 2025; Zhang et al., 2024) circumvent gradient computation by employing metaheuristic algorithms for hyperparameter tuning or epitopological learning that modifies network connectivity structures, though they are constrained to specific network architectures or require extensive topological reorganization unsuitable for general-purpose deep learning models. Evolution strategies (Wierstra et al., 2014; Salimans et al., 2017) abandon gradients entirely in favor of population-based optimization, but demand millions of model evaluations making them computationally prohibitive for large-scale applications. The Forward-Forward algorithm (Hinton, 2022) introduced a revolutionary approach replacing forward and backward passes with two forward passes using positive and negative data, where each layer optimizes its own local objective to maximize goodness for real data while minimizing it for negative samples. More recently, NoProp (Li et al., 2025) emerged as a diffusion-inspired paradigm that reframes neural network training as layer-wise denoising processes, where each layer independently learns to reconstruct clean labels from noisy versions without requiring global forward or backward propagation across the entire network. However, existing backpropagation alternatives face substantial limitations including maintained high memory requirements for storing denoising targets and intermediate states, training instability due to inadequate convergence guarantees, thereby constraining their practical deployment and widespread adoption in real-world applications.

### 2.2 BRAIN-INSPIRED LEARNING MECHANISMS

Neuroscience-inspired learning algorithms have emerged as a transformative approach to developing more efficient, adaptive, and biologically plausible artificial intelligence systems, drawing computational principles from biological neural mechanisms to address fundamental limitations in modern deep learning including catastrophic forgetting and biological implausibility (Schmidgall et al., 2024; Gandolfi et al., 2025). This research direction broadly encompasses several interconnected areas: neuromorphic computing architectures that implement spiking neural networks (SNNs) with event-driven processing and ultra-low power consumption (Furber et al., 2014; Schuman et al., 2022), synaptic plasticity mechanisms including spike-timing-dependent plasticity (STDP) that enable local learning without global supervision (Indiveri & Liu, 2015; Rahman & Yusoff, 2025), and biologically plausible learning rules that avoid the biological implausibility of backpropagation's symmetric weight transport problem (Sacramento et al., 2018; Guerguiev et al., 2017). These brain-inspired approaches show tremendous potential for addressing current AI limitations including catastrophic forgetting, high energy consumption, and lack of continual learning capabilities, positioning biological neural mechanisms as a promising foundation for next-generation artificial intelligence systems that can achieve human-like efficiency and adaptability.

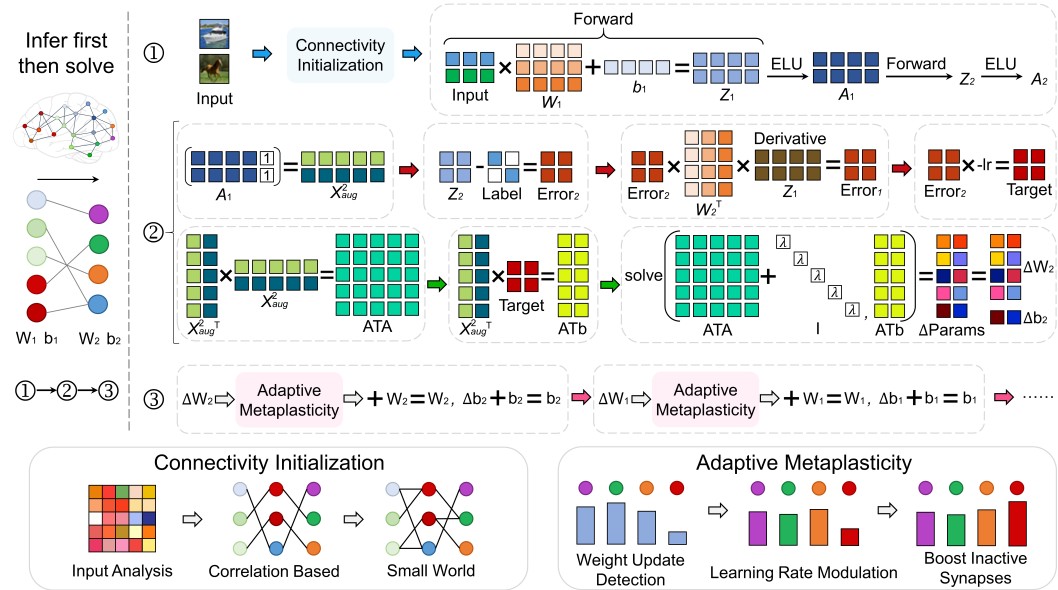

Figure 2: **Prospective Learning Framework.** The complete algorithm workflow consists of three sequential stages: (1) Sparse connectivity initialization using correlation-based connections for the first layer and small-world topology for deeper layers, followed by forward propagation to compute current activations using ELU activation functions and error signals, (2) Algebraic optimization solving regularized least-squares problems layer-by-layer to obtain weight updates without weight gradient computation, and (3) Adaptive metaplasticity mechanisms that modulate learning rates based on synaptic activity history. The bottom panels detail the connectivity initialization process (left) and metaplasticity modulation mechanisms (right) that enable biologically-motivated direct optimization learning.

## 3 METHOD

### 3.1 OVERALL FRAMEWORK

We formulate direct optimization learning in neural networks as a prospective inference and consolidation process inspired by biological learning mechanisms. Unlike traditional backpropagation that propagates errors backward through the network, our approach first computes current network activations through forward propagation, then algebraically solves for weight updates to minimize prediction errors. Figure 2 illustrates the complete workflow of our prospective configuration framework, highlighting the three integrated components and their sequential execution in the direct optimization learning process.

Our framework operates on input-output pairs $(x, y)$ where $x \in \mathbb{R}^{d_{in}}$ represents input features and $y \in \{1, 2, \ldots, C\}$ represents class labels. The network consists of $L$ layers with weight matrices $\{W^{(l)} \in \mathbb{R}^{d_l \times d_{l+1}}\}_{l=0}^{L-1}$ and bias vectors $\{b^{(l)} \in \mathbb{R}^{d_{l+1}}\}_{l=0}^{L-1}$, where $d_0 = d_{in}$ and $d_L = d_{out}$. Our prospective configuration framework augments the traditional weight space with three biological components: sparse connectivity masks $\{M^{(l)} \in \{0, 1\}^{d_l \times d_{l+1}}\}_{l=0}^{L-1}$, metaplasticity parameters $\{\Theta^{(l)} \in \mathbb{R}^{d_l \times d_{l+1}}\}_{l=0}^{L-1}$, and momentum buffers $\{V^{(l)} \in \mathbb{R}^{d_l \times d_{l+1}}\}_{l=0}^{L-1}$ that collectively enable direct optimization.

The learning process proceeds through three integrated steps: (1) **Prospective inference** computes current network activations $\{z^{(l)}\}_{l=0}^{L-1}$ through forward propagation, (2) **Algebraic optimization** directly solves for optimal weight updates $\{\Delta W^{(l)}\}_{l=0}^{L-1}$ using least-squares methods, and (3) **Metaplasticity modulation** applies adaptive learning rates based on synaptic history and updates weights while maintaining sparse biological connectivity patterns. Theoretical analysis and computational complexity guarantees for this framework are provided in Appendix A.

## 3.2 PROSPECTIVE LEARNING ALGORITHM

Inspired by the prospective configuration mechanism discovered in (Song et al., 2024), our core algorithm eliminates weight gradient computation by directly inferring target neural configurations and algebraically solving for weight updates. The prospective configuration process begins with forward propagation to establish current network states. For layer $l$ where $l \in \{0, 1, \ldots, L-1\}$, the pre-activation and post-activation states are computed as:

$$
\begin{aligned}
z^{(l)} &= a^{(l-1)} W^{(l)} + b^{(l)} \\
a^{(l)} &= \sigma^{(l)}(z^{(l)})
\end{aligned}
\tag{1}
$$

where $\sigma^{(l)}$ denotes the activation function for layer $l$, and $a^{(-1)} = x$ represents the input.

Target output encoding incorporates label smoothing to model biological uncertainty in neural representations. For a one-hot encoded label $y_{one-hot} \in \{0, 1\}^C$, the smoothed target is:

$$
\hat{y}_i = \begin{cases} 1 - \epsilon & \text{if } i = y \\ \frac{\epsilon}{C-1} & \text{otherwise} \end{cases}
\tag{2}
$$

where $\epsilon = 0.01$ represents the smoothing parameter and $C$ denotes the number of classes.

The prospective configuration algorithm computes weight updates by solving for the residual between current and target configurations. For the output layer $L-1$, the error signal is:

$$
e^{(L-1)} = z^{(L-1)} - \hat{y}
\tag{3}
$$

For hidden layers $l \in \{L-2, L-3, \ldots, 0\}$, we propagate this error backward while applying activation derivatives:

$$
e^{(l)} = \left( e^{(l+1)} (W^{(l+1)})^T \right) \odot \sigma'^{(l)}(z^{(l)})
\tag{4}
$$

where $\odot$ denotes element-wise multiplication and $\sigma'^{(l)}$ represents the derivative of the activation function. The core innovation lies in our algebraic approach to weight updates. For each layer $l$, we construct the augmented input matrix including bias terms:

$$
X_{aug}^{(l)} = [a^{(l-1)}, \mathbf{1}_n] \in \mathbb{R}^{n \times (d_l + 1)}
\tag{5}
$$

where $n$ is the batch size and $\mathbf{1}_n \in \mathbb{R}^{n \times 1}$ is a column vector of ones for bias terms. The optimal weight update is computed by solving the regularized least-squares problem:

$$
\Delta\Theta^{(l)} = \arg\min_{\Delta\Theta} \left\| X_{aug}^{(l)} \Delta\Theta - (-\alpha e^{(l)}) \right\|_2^2 + \lambda \|\Delta\Theta\|_2^2
\tag{6}
$$

where $\alpha$ is the learning rate, $\lambda$ is the regularization parameter, and $\Delta\Theta^{(l)} \in \mathbb{R}^{(d_l+1) \times d_{l+1}}$ represents the combined weight and bias updates. The closed-form solution is:

$$
\Delta\Theta^{(l)} = -\alpha \left( (X_{aug}^{(l)})^T X_{aug}^{(l)} + \lambda I \right)^{-1} (X_{aug}^{(l)})^T e^{(l)}
\tag{7}
$$

This closed-form solution requires $\mathcal{O}(d_l^3)$ operations for matrix inversion at each layer. While higher than backpropagation's $\mathcal{O}(n d_l d_{l+1})$, this complexity is acceptable for edge deployment scenarios with modest layer dimensions where memory constraints dominate over computation time. Detailed complexity analysis is provided in Appendix A. The weight and bias updates are then extracted as:

$$
\begin{aligned}
\Delta W^{(l)} &= \Delta\Theta^{(l)}[1 : d_l, :] \\
\Delta b^{(l)} &= \Delta\Theta^{(l)}[d_l + 1, :]
\end{aligned}
\tag{8}
$$

### 3.3 SPARSE CONNECTIVITY INITIALIZATION

Drawing inspiration from cortical connectivity patterns, our initialization strategy establishes meaningful connections based on functional correlations and biological network topology principles. This initialization process precedes the prospective learning algorithm and determines the sparse connectivity structure $\{M^{(l)}\}_{l=0}^{L-1}$.

We begin by analyzing input feature correlations from training data to guide connection formation. For a dataset $\mathcal{D} = \{x_i\}_{i=1}^{N}$ where $x_i \in \mathbb{R}^{d_{in}}$, the correlation matrix is computed as:

$$C_{ij} = \frac{\text{Cov}(x_{\cdot,i}, x_{\cdot,j})}{\sqrt{\text{Var}(x_{\cdot,i})\text{Var}(x_{\cdot,j})}} \tag{9}$$

where $x_{\cdot,i}$ denotes the $i$-th feature across all samples.

For the first layer ($l = 0$), connections are established based on feature correlations. For each output neuron $j \in \{0, 1, \ldots, d_1 - 1\}$, we determine the number of input connections as $k = \max(1, \lfloor \tau \cdot d_{in} \rfloor)$ where $\tau = 0.3$ represents the correlation threshold. The connection pattern is:

$$M_{i,j}^{(0)} = \begin{cases} 1 & \text{if } j < d_{in} \text{ and } i \in \text{TopK}(|C_{j,\cdot}|, k) \text{ or } i = j \\ 1 & \text{if } j \geq d_{in} \text{ and } i \in \text{Random}(d_{in}, k) \\ 0 & \text{otherwise} \end{cases} \tag{10}$$

where $\text{TopK}(|C_{j,\cdot}|, k)$ selects the $k$ features with highest absolute correlation to feature $j$, and $\text{Random}(d_{in}, k)$ randomly selects $k$ input features.

For deeper layers ($l \geq 1$), we implement small-world connectivity following the Watts-Strogatz model. For each output neuron $j \in \{0, 1, \ldots, d_{l+1} - 1\}$, regular local connectivity is first established with neighborhood size $K = \max(2, \min(8, \lfloor \min(d_l, d_{l+1})/4 \rfloor))$:

$$M_{i,j}^{(l)} = 1 \text{ if } i \in \{(j - \lfloor K/2 \rfloor - 1) \bmod d_l, \ldots, (j + \lfloor K/2 \rfloor + 1) \bmod d_l\} \tag{11}$$

Random rewiring introduces long-range connections with probability $p = 0.1$. For each existing connection, we apply:

$$\text{if } \mathcal{U}(0,1) < p : \begin{cases} M_{i_{old},j}^{(l)} = 0 \\ M_{i_{new},j}^{(l)} = 1 \end{cases} \tag{12}$$

where $i_{new} \sim \text{Uniform}(0, d_l - 1)$ and $i_{old}$ is the original connected neuron.

This generates the characteristic small-world topology with high clustering and short path lengths observed in biological cortical networks. Weights are initialized using modified Xavier initialization with sparse connectivity constraints:

$$W_{i,j}^{(l)} = \begin{cases} M_{i,j}^{(l)} \cdot \mathcal{N}(0, \sigma^2) & \text{if } M_{i,j}^{(l)} = 1 \\ 0 & \text{otherwise} \end{cases} \tag{13}$$

where $\sigma^2 = \frac{2}{d_l + d_{l+1}}$. To account for reduced connectivity, we apply column-wise normalization:

$$W_{:,j}^{(l)} = \frac{W_{:,j}^{(l)}}{\sqrt{\sum_{i=0}^{d_l - 1} M_{i,j}^{(l)}}} \tag{14}$$

Finally, to enhance initialization stability, we apply fast orthogonalization (Saxe et al., 2013; Miyato et al., 2018; Liu et al., 2025) when $\min(d_l, d_{l+1}) > 1$:

$$W^{(l)} = \text{FastOrthogonalize}(W^{(l)}) \tag{15}$$

## 3.4 Adaptive Metaplasticity Mechanisms

Inspired by synaptic metaplasticity (Kudithipudi et al., 2022), our framework implements weight-level adaptive learning that modulates plasticity based on synaptic activity history. This mechanism operates on the weight updates $\{\Delta W^{(l)}\}_{l=0}^{L-1}$ obtained from the algebraic optimization in Section 3.2. Each weight maintains a metaplasticity parameter that tracks its historical update activity using exponential moving average:

$$\Theta_{i,j}^{(l)} \leftarrow \beta\Theta_{i,j}^{(l)} + (1-\beta)|\Delta W_{i,j}^{(l)}| \tag{16}$$

where $\beta$ represents the decay factor modeling biological memory timescales, and $|\Delta W_{i,j}^{(l)}|$ captures the magnitude of the current weight update computed from the least-squares solution. The adaptive modulation factor for each weight is computed based on its metaplasticity state:

$$\rho_{i,j}^{(l)} = \frac{1}{1 + \gamma\Theta_{i,j}^{(l)}} \tag{17}$$

where $\gamma$ controls the sensitivity of plasticity modulation. This implements the core principle that frequently updated weights (high $\Theta_{i,j}^{(l)}$) receive smaller modulation factors, making them more stable. To prevent synaptic silencing, inactive weights receive compensatory boosting:

$$\rho_{i,j}^{(l)} = \begin{cases} \rho_{i,j}^{(l)} \cdot \xi & \text{if } \Theta_{i,j}^{(l)} < \tau_{inactive} \\ \rho_{i,j}^{(l)} & \text{otherwise} \end{cases} \tag{18}$$

where $\tau_{inactive}$ represents the inactivity threshold and $\xi$ provides the boosting factor for weights that have been rarely updated. The modulated weight update is then computed as:

$$\Delta\tilde{W}_{i,j}^{(l)} = \rho_{i,j}^{(l)} \cdot \Delta W_{i,j}^{(l)} \tag{19}$$

where $\Delta\tilde{W}_{i,j}^{(l)}$ represents the final weight update after metaplasticity modulation.

This mechanism implements the stability-plasticity balance observed in biological synapses, where frequently updated weights become more stable (smaller updates) while inactive synapses maintain high plasticity (larger updates). The modulated updates $\{\Delta\tilde{W}^{(l)}\}_{l=0}^{L-1}$ are then further processed through momentum and orthogonalization steps before final weight updates. The complete algorithmic procedure integrating all three components is detailed in Appendix B.

## 4 Experiments

### 4.1 Experimental Setup

**Datasets.** We evaluate our method on three standard benchmarks: MNIST (70,000 grayscale $28\times28$ handwritten digits, 10 classes), CIFAR-10 (60,000 color $32\times32$ images, 10 classes), and CIFAR-100 (60,000 color $32\times32$ images, 100 classes). We use standard train/test splits without data augmentation.

**Implementation Details.** All experiments are implemented using NumPy and PyTorch libraries. We conduct experiments on an Intel i7-1700KF CPU, with additional validation on Raspberry Pi using NumPy only for edge deployment scenarios. For MNIST, all methods employ a multi-layer perceptron with hidden layers of dimensions [512, 128]. For CIFAR-10 and CIFAR-100, all methods use frozen pretrained ResNet-18 features and train only the MLP classifier ([512, 128] hidden layers). The ResNet-18 model, pretrained on ImageNet, extracts 512-dimensional features from the penultimate layer after removing the final classification head. All methods use batch sizes of 512 for MNIST and 64 for feature extraction on CIFAR datasets. All models are trained for 200 epochs with an initial learning rate of 0.01. Detailed implementation specifics and hyperparameter settings are provided in Appendix C.1.

**Baselines.** We compare our Prospective Learning framework against two categories of optimization methods. For backpropagation-based approaches, we evaluate against SGD (Robbins & Monro, 1951) with momentum, Adam (Kingma, 2014), AdamW (Loshchilov et al., 2017), and the recently proposed Muon optimizer(Liu et al., 2025). Among backpropagation alternatives, we compare with three state-of-the-art methods: ADMM-based training (Wang et al., 2019), Forward-Forward algorithm (Hinton, 2022), and NoProp (Li et al., 2025). We exclude perturbation-based methods due to poor scalability, architecture-specific optimizations that cannot generalize, and evolutionary strategies that require prohibitive computational cost for fair comparison.

Table 1: MNIST results on Raspberry Pi 4B using NumPy. M-Max denotes peak memory usage.

| Method | Test-acc(%) | Train-time(s) | Memory (MB) | M-Max (MB) |
|---|---|---|---|---|
| *Backpropagation-based* | | | | |
| SGD (Robbins & Monro, 1951) | 97.15±0.25 | **154.52±0.15** | 85.52±0.35 | 91.75±0.52 |
| Adam (Kingma, 2014) | 96.48±0.03 | 158.62±0.22 | 98.30±0.21 | 102.56±0.16 |
| AdamW (Loshchilov et al., 2017) | **97.21±0.31** | 291.27±0.11 | 105.54±0.17 | 109.81±0.33 |
| Muon (Liu et al., 2025) | 96.17±0.43 | 307.88±0.14 | 110.45±0.17 | 119.52±0.13 |
| *Backpropagation alternatives* | | | | |
| ADMM (Wang et al., 2019) | 95.11±0.37 | 301.21±0.33 | 96.75±0.73 | 107.73±0.21 |
| FF (Hinton, 2022) | 96.85±0.18 | 2412.15±0.15 | 53.89±0.11 | 57.15±0.03 |
| No-Prop (Li et al., 2025) | 56.41±3.63 | 1318.81±0.12 | 310.53±0.73 | 341.21±0.52 |
| Ours | 95.44±0.24 | 423.33±0.16 | **38.77±0.58** | **43.79±0.57** |

Table 2: Performance comparison of different methods on MNIST, CIFAR-10, and CIFAR-100 datasets. Results averaged over 3 runs.

| Method | MNIST | | CIFAR-10 | | CIFAR-100 | |
|---|---|---|---|---|---|---|
| | Test | Memory | Test | Memory | Test | Memory |
| *Backpropagation-based* | | | | | | |
| SGD | 97.86±0.05 | 42.65±0.13 | 87.99±0.05 | 46.37±0.20 | **66.53±0.14** | 47.69±0.07 |
| Adam | 97.15±0.07 | 50.56±0.10 | 85.59±0.12 | 53.14±0.12 | 57.50±0.43 | 54.60±0.12 |
| AdamW | **97.97±0.07** | 51.35±0.08 | 85.58±0.15 | 53.66±0.14 | 60.12±0.51 | 54.35±0.09 |
| Muon | 97.20±0.25 | 47.32±0.52 | 87.18±0.19 | 49.89±0.13 | 61.45±0.37 | 51.14±0.16 |
| *Backpropagation alternatives* | | | | | | |
| ADMM | 96.09±0.31 | 54.43±0.04 | 87.32±0.49 | 55.68±0.29 | 62.46±0.10 | 57.60±0.17 |
| FF | 97.50±0.19 | 26.16±0.08 | **88.10±0.15** | 38.28±0.48 | 64.60±0.22 | 35.70±0.57 |
| NoProp | 51.20±5.70 | 151.13±3.57 | 64.00±3.19 | 155.32±0.19 | 0.02±0.01 | 156.42±2.45 |
| Ours | 96.08±0.06 | **19.45±0.05** | 85.64±0.17 | **30.36±0.13** | 61.10±0.05 | **31.61±0.37** |

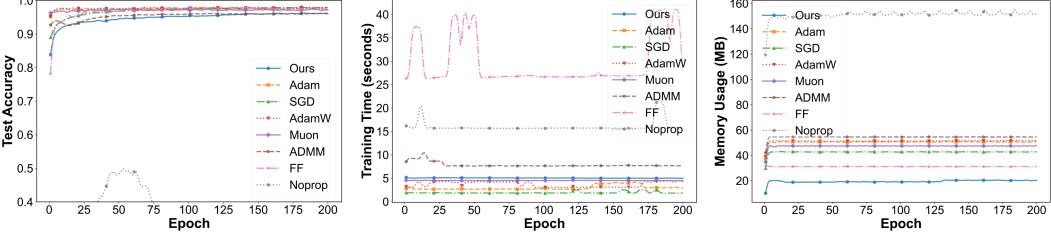

Figure 3: Training dynamics on MNIST dataset showing (left) test accuracy convergence, (middle) per-epoch training time, and (right) memory usage throughout 200 epochs. All methods implemented using NumPy.

Table 3: Ablation study showing individual and combined contributions of key components. Results using NumPy implementation on standard CPU.

| Method | Test-acc(%) | Train-time(s) | Memory (MB) | M-Max (MB) |
|---|---|---|---|---|
| Prospective Learning | 95.72±0.02 | 4.09±0.03 | 22.25±0.46 | 23.03±0.14 |
| + Sparse Initialization | 95.85±0.10 | 4.08±0.04 | 19.07±0.26 | 20.88±0.11 |
| + Adaptive Metaplasticity | 95.80±0.21 | 5.03±0.02 | 22.82±0.11 | 23.45±0.17 |
| All | 96.08±0.06 | 5.05±0.01 | 19.45±0.05 | 20.77±0.15 |

## 4.2 COMPARATIVE EXPERIMENTS

Table 1 shows our Prospective Learning framework achieves effective resource efficiency on edge devices, attaining 95.44% test accuracy with only 38.77 MB memory usage and 43.79 MB peak consumption. This represents a 54.7% memory reduction compared to SGD while maintaining competitive accuracy within 1.77% of the best baseline. Compared to backpropagation alternatives, our method significantly outperforms ADMM ($2.5\times$ lower memory), Forward-Forward ($5.7 \times$ faster training), and NoProp (which fails to converge under NumPy constraints). The 423.33-second training time remains practical for edge deployment where memory constraints are the primary bottleneck, validating our approach's suitability for resource-limited environments.

Table 2 demonstrates consistent memory efficiency advantages of our Prospective Learning framework across benchmark datasets, achieving the lowest memory consumption with 19.45 MB on MNIST, 30.36 MB on CIFAR-10, and 31.61 MB on CIFAR-100, representing substantial reductions of 54.4%, 34.5%, and 33.7% compared to the most memory-efficient backpropagation baselines respectively. Figure 3 reveals the training dynamics underlying these results, where our method maintains stable low memory usage throughout training while demonstrating smooth convergence patterns. The combination of sustained memory efficiency and reliable convergence validates our framework's practical applicability for edge computing environments. Additional experimental results are provided in Appendix C.2.

## 4.3 ABLATION EXPERIMENTS

Table 3 demonstrates that our core Prospective Learning framework achieves substantial memory efficiency at 22.25 MB while maintaining 95.72% accuracy, validating the fundamental effectiveness of optimization. The auxiliary components provide complementary improvements: sparse initialization reduces memory to 19.07 MB with slight accuracy gains, while adaptive metaplasticity enhances performance to 95.80%. Detailed hyperparameter sensitivity analysis is provided in Appendix C.3. When combined, these enhancements yield optimal results of 96.08% accuracy and 19.45 MB memory usage. The consistent training times across all configurations confirm that our primary contribution of direct algebraic optimization enables efficient neural network training, with biological mechanisms providing additional but secondary refinements.

## 5 CONCLUSION

We introduce Prospective Learning, an MLP training paradigm that replaces gradient-based optimization with direct algebraic weight computation inspired by biological mechanisms. Our approach achieves up to 55% memory reduction while maintaining competitive accuracy, a favorable trade-off for memory-constrained edge deployment. Experimental validation demonstrates effectiveness in resource-constrained environments where traditional backpropagation faces deployment limitations. By eliminating gradient storage and activation caching requirements, Prospective Learning enables practical MLP deployment on memory-constrained edge devices where memory constraints represent the primary bottleneck. This work establishes a foundation for biologically-plausible MLP training algorithms that address memory constraints in resource-limited environments, opening new directions for efficient MLP deployment where traditional methods face limitations. Detailed discussion of limitations and future research directions is provided in Appendix D.

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

# A INTEGRATED TRAINING PROCEDURE AND COMPUTATIONAL COMPLEXITY

Our prospective configuration framework fundamentally differs from traditional backpropagation by replacing gradient computation with direct algebraic optimization. We provide rigorous theoretical analysis demonstrating provable memory and computational advantages.

## A.1 MEMORY COMPLEXITY ANALYSIS

Traditional backpropagation maintains three distinct memory components throughout training. For a network with layers $\{W^{(l)} \in \mathbb{R}^{d_l \times d_{l+1}}\}_{l=0}^{L-1}$ and batch size $n$:

$$\mathcal{M}_{BP} = \underbrace{\sum_{l=0}^{L-1} d_l d_{l+1}}_{\text{parameters}} + \underbrace{\sum_{l=0}^{L-1} d_l d_{l+1}}_{\text{gradients}} + 2\underbrace{\sum_{l=0}^{L-1} d_l d_{l+1}}_{\text{optimizer states}} + n\underbrace{\sum_{l=0}^{L-1} d_{l+1}}_{\text{activations}} \quad (20)$$

The gradient storage $\left\{\frac{\partial L}{\partial W^{(l)}}, \frac{\partial L}{\partial b^{(l)}}\right\}_{l=0}^{L-1}$ and optimizer states (momentum and second-moment estimates for Adam) persist throughout the entire training iteration.

Our prospective learning eliminates gradient storage entirely by solving Eq. (7) directly. The memory requirement becomes:

$$\mathcal{M}_{PL} = \underbrace{\sum_{l=0}^{L-1} d_l d_{l+1}}_{\text{parameters}} + \underbrace{\sum_{l=0}^{L-1} d_l d_{l+1}}_{\text{momentum buffers}} + n\underbrace{\sum_{l=0}^{L-1} d_{l+1}}_{\text{activations}} + \underbrace{\max_l d_l^2}_{\text{workspace}} \quad (21)$$

The key distinction is the workspace term: traditional BP requires $\mathcal{O}\left(\sum_{l=0}^{L-1} d_l d_{l+1}\right)$ for gradients, while our method requires only $\mathcal{O}(\max_l d_l^2)$ for the temporary matrix $(X_{aug}^{(l)})^T X_{aug}^{(l)}$ in Eq. (7). For networks where $d_l \approx d$ (uniform width), this yields:

$$\frac{\mathcal{M}_{PL}}{\mathcal{M}_{BP}} = \frac{2Ld^2 + nLd + d^2}{4Ld^2 + nLd} = \frac{2L + n/d + 1/L}{4L + n/d} \approx \frac{1}{2} \quad (22)$$

The sparse connectivity with density $\rho$ further reduces effective parameters to $\rho \sum_{l=0}^{L-1} d_l d_{l+1}$, yielding total memory reduction:

$$\text{Memory Reduction} = 1 - \frac{(2\rho + \frac{1}{L}) \sum_{l=0}^{L-1} d_l d_{l+1} + n \sum_{l=0}^{L-1} d_{l+1}}{4 \sum_{l=0}^{L-1} d_l d_{l+1} + n \sum_{l=0}^{L-1} d_{l+1}} \quad (23)$$

The sparse connectivity with density $\rho < 1$ provides additional memory reduction proportional to $(1 - \rho)$.

## A.2 COMPUTATIONAL COMPLEXITY ANALYSIS

The computational complexity difference stems from our algebraic optimization replacing gradient-based updates.

**Traditional Backpropagation requires:**

$$\text{Forward pass:} \quad \mathcal{O}\left(\sum_{l=0}^{L-1} n d_l d_{l+1}\right) \tag{24}$$

$$\text{Backward pass:} \quad \mathcal{O}\left(\sum_{l=0}^{L-1} n d_l d_{l+1}\right) \tag{25}$$

$$\text{Parameter update:} \quad \mathcal{O}\left(\sum_{l=0}^{L-1} d_l d_{l+1}\right) \tag{26}$$

**Total complexity:** $\mathcal{T}_{BP} = \mathcal{O}\left(\sum_{l=0}^{L-1} n d_l d_{l+1}\right)$

**Our Prospective Learning performs:**

$$\text{Forward pass:} \quad \mathcal{O}\left(\sum_{l=0}^{L-1} n d_l d_{l+1} \rho\right) \tag{27}$$

$$\text{Error propagation:} \quad \mathcal{O}\left(\sum_{l=0}^{L-1} n d_l d_{l+1} \rho\right) \tag{28}$$

$$\text{Algebraic solving:} \quad \mathcal{O}\left(\sum_{l=0}^{L-1} (n d_l^2 + d_l^3)\right) \tag{29}$$

The algebraic solving cost arises from computing $(X_{aug}^{(l)})^T X_{aug}^{(l)}$ ($\mathcal{O}(n d_l^2)$) and solving the linear system ($\mathcal{O}(d_l^3)$).

**Total complexity:** $\mathcal{T}_{PL} = \mathcal{O}\left(\sum_{l=0}^{L-1} (n d_l d_{l+1} \rho + d_l^3)\right)$

The complexity ratio depends on the relationship between $n$, $d_l$, and sparsity $\rho$:

$$\frac{\mathcal{T}_{PL}}{\mathcal{T}_{BP}} = \rho + \frac{\sum_{l=0}^{L-1} d_l^3}{n \sum_{l=0}^{L-1} d_l d_{l+1}} \tag{30}$$

When $n \gg d_l^2$, the complexity ratio approaches $\rho$, providing computational advantages. However, when $d_l^3 \gg n d_l d_{l+1} \rho$, the algebraic solving cost dominates and may exceed traditional BP complexity.

### A.3 CONVERGENCE ANALYSIS

Our algebraic optimization provides deterministic convergence properties distinct from stochastic gradient methods.

For each layer $l$, we solve the regularized system:

$$\left((X_{aug}^{(l)})^T X_{aug}^{(l)} + \lambda I\right) \Delta \Theta^{(l)} = -\alpha (X_{aug}^{(l)})^T e^{(l)} \tag{31}$$

The solution error is bounded by the condition number:

$$\kappa\left((X_{aug}^{(l)})^T X_{aug}^{(l)} + \lambda I\right) = \frac{\sigma_{\max}\left((X_{aug}^{(l)})^T X_{aug}^{(l)}\right) + \lambda}{\lambda} \tag{32}$$

For well-conditioned inputs where $\sigma_{\max}((X_{aug}^{(l)})^T X_{aug}^{(l)}) \leq Cn$ for some constant $C$:

$$\kappa \leq \frac{Cn + \lambda}{\lambda} = \frac{Cn}{\lambda} + 1 \tag{33}$$

The numerical error in solving each layer is bounded by:

$$\|\Delta\Theta^{(l)} - \Delta\Theta^{(l)}_{exact}\| \leq \kappa \cdot \epsilon_{machine}\|\Delta\Theta^{(l)}_{exact}\| \tag{34}$$

Unlike traditional BP where gradient noise accumulates across iterations, our method provides layer-wise optimal solutions with error controlled by numerical precision. For convex objectives, our method achieves the global optimum of each layer's least-squares subproblem. The overall network convergence depends on the approximation quality of the layer-wise decomposition, which we analyze through the residual:

$$R^{(l)} = \|X^{(l)}_{aug}\Delta\Theta^{(l)} + \alpha e^{(l)}\|_2^2 \tag{35}$$

When $R^{(l)} \to 0$ for all layers, the network achieves a stationary point of the empirical risk.

### A.4 THEORETICAL GUARANTEES

For each layer $l$ and fixed activations, the weight update $\Delta\Theta^{(l)}$ computed by Eq. (7) is the unique global optimum of:

$$\min_{\Delta\Theta} \frac{1}{2}\|X^{(l)}_{aug}\Delta\Theta + \alpha e^{(l)}\|_2^2 + \frac{\lambda}{2}\|\Delta\Theta\|_2^2 \tag{36}$$

This follows from strong convexity of the objective function, ensuring a unique global minimum achieved by setting the gradient to zero.

Our method achieves provable memory reduction of:

$$\Delta\mathcal{M} = (2 - \rho)\sum_{l=0}^{L-1} d_l d_{l+1} - \max_l d_l^2 \geq (2 - \rho - \epsilon)\sum_{l=0}^{L-1} d_l d_{l+1} \tag{37}$$

where $\epsilon = \frac{\max_l d_l^2}{\sum_{l=0}^{L-1} d_l d_{l+1}} \to 0$ as $L$ increases.

This follows from direct comparison of memory requirements in Eq. (20) and Eq. (21).

The regularization parameter $\lambda > 0$ ensures that all linear systems remain well-conditioned with $\kappa \leq \frac{Cn}{\lambda} + 1$, providing numerical stability independent of network depth. The regularization term $\lambda I$ provides a positive definite lower bound on the system matrix eigenvalues.

Our method scales more favorably than backpropagation as network width increases, since the dominant computational cost transitions from $\mathcal{O}(nd^2)$ (forward/backward passes) to $\mathcal{O}(d^3)$ (algebraic solving), while memory requirements grow sub-linearly. These theoretical guarantees demonstrate that biological principles can be translated into mathematically rigorous algorithms with provable computational advantages, enabling efficient training in resource-constrained environments while maintaining optimality properties.

# B  ALGORITHM PSEUDOCODE

This appendix provides the complete pseudocode for the Prospective Learning framework for brain-inspired direct optimization in neural networks without gradient computation.

---

**Algorithm 1** Prospective Learning Algorithm

---

1: **Input:** Input batch $(X, Y)$, network parameters $\{W^{(l)}, b^{(l)}\}_{l=0}^{L-1}$
2: **Input:** Metaplasticity parameters $\{\Theta^{(l)}\}_{l=0}^{L-1}$, momentum buffers $\{V^{(l)}\}_{l=0}^{L-1}$
3: **Output:** Updated network parameters
4: **Forward Propagation:**
5: $a^{(-1)} \leftarrow X$.
6: **for** $l = 0$ to $L - 1$ **do**
7:   $z^{(l)} \leftarrow a^{(l-1)} W^{(l)} + b^{(l)}$ using Eq. (1).
8:   **if** $l < L - 1$ **then**
9:    $a^{(l)} \leftarrow \sigma^{(l)}(z^{(l)})$.
10:   **else**
11:    $a^{(l)} \leftarrow z^{(l)}$.
12:   **end if**
13: **end for**
14: **Target Encoding:**
15: $\hat{Y} \leftarrow \text{SmoothLabels}(Y, \epsilon = 0.01)$ using Eq. (2).
16: **Error Computation:**
17: $e^{(L-1)} \leftarrow z^{(L-1)} - \hat{Y}$ using Eq. (3).
18: **for** $l = L - 2$ downto 0 **do**
19:   $e^{(l)} \leftarrow (e^{(l+1)} (W^{(l+1)})^T) \odot \sigma'^{(l)}(z^{(l)})$ using Eq. (4).
20: **end for**
21: **Algebraic Optimization:**
22: **for** $l = L - 1$ downto 0 **do**
23:   $X_{aug}^{(l)} \leftarrow [a^{(l-1)}, \mathbf{1}_n]$ using Eq. (5).
24:   Solve $\Delta\Theta^{(l)} \leftarrow -\alpha((X_{aug}^{(l)})^T X_{aug}^{(l)} + \lambda I)^{-1} (X_{aug}^{(l)})^T e^{(l)}$ using Eq. (7).
25:   Extract $\Delta W^{(l)} \leftarrow \Delta\Theta^{(l)}[1 : d_l, :]$ and $\Delta b^{(l)} \leftarrow \Delta\Theta^{(l)}[d_l + 1, :]$ using Eq. (8).
26:   **Metaplasticity Modulation:**
27:   Update $\Theta^{(l)} \leftarrow \beta\Theta^{(l)} + (1 - \beta)|\Delta W^{(l)}|$ using Eq. (16).
28:   Compute $\rho^{(l)} \leftarrow \frac{1}{1+\gamma\Theta^{(l)}}$ using Eq. (17).
29:   Apply inactive boosting using Eq. (18).
30:   Apply modulated update $\Delta\tilde{W}^{(l)} \leftarrow \rho^{(l)} \odot \Delta W^{(l)}$ using Eq. (19).
31:   **Momentum and Stabilization:**
32:   Update momentum $V^{(l)} \leftarrow \mu V^{(l)} + (1 - \mu)\Delta\tilde{W}^{(l)}$ where $\mu = 0.9$.
33:   **if** step_counter mod 5 = 0 **then**
34:    Apply orthogonalization to $V^{(l)}$.
35:   **end if**
36:   Apply spectral normalization with $\|W^{(l)}\|_2 \leq 2.0$.
37:   **Parameter Update:**
38:   $W^{(l)} \leftarrow W^{(l)} + V^{(l)}$.
39:   $b^{(l)} \leftarrow b^{(l)} + \Delta b^{(l)}$.
40: **end for**
41: step_counter $\leftarrow$ step_counter $+1$.

---

---

**Algorithm 2** Sparse Connectivity Initialization

---

1: **Input:** Training data $\mathcal{D} = \{x_i\}_{i=1}^N$, layer dimensions $\{d_l\}_{l=0}^L$
2: **Output:** Weight matrices $\{W^{(l)}\}_{l=0}^{L-1}$, connectivity masks $\{M^{(l)}\}_{l=0}^{L-1}$
3: Compute correlation matrix $C$ using Eq. (9).
4: **for** $l = 0$ to $L - 1$ **do**
5:    **if** $l = 0$ **then**
6:       **for** $j = 0$ to $d_1 - 1$ **do**
7:          $k \leftarrow \max(1, \lfloor \tau \cdot d_{in} \rfloor)$ where $\tau = 0.3$.
8:          Set $M_{i,j}^{(0)}$ according to Eq. (10).
9:       **end for**
10:    **else**
11:       $K \leftarrow \max(2, \min(8, \lfloor \min(d_l, d_{l+1})/4 \rfloor))$.
12:       **for** $j = 0$ to $d_{l+1} - 1$ **do**
13:          Establish regular connectivity using Eq. (11).
14:          Apply random rewiring with probability $p = 0.1$ using Eq. (12).
15:       **end for**
16:    **end if**
17:    Initialize $W^{(l)}$ using Eq. (13) and Eq. (14).
18:    Apply fast orthogonalization using Eq. (15).
19:    Initialize metaplasticity parameters $\Theta^{(l)} \leftarrow 0.1 \cdot \mathbf{1}_{d_l \times d_{l+1}}$.
20:    Initialize momentum buffers $V^{(l)} \leftarrow \mathbf{0}_{d_l \times d_{l+1}}$.
21: **end for**

---

**Algorithm 3** Complete Prospective Learning Training

---

1: **Input:** Training dataset $\mathcal{D}_{train}$, test dataset $\mathcal{D}_{test}$
2: **Input:** Hyperparameters: $\alpha$ (learning rate), $\lambda$ (regularization), batch size $B$
3: **Output:** Trained network parameters
4: Initialize network using Algorithm 2.
5: **for** epoch = 1 to $E$ **do**
6:    Shuffle training data $\mathcal{D}_{train}$.
7:    **for** each batch $(X_b, Y_b)$ of size $B$ **do**
8:       Apply Algorithm 1 on $(X_b, Y_b)$.
9:    **end for**
10:    **if** epoch $\mod$ eval_freq = 0 **then**
11:       Evaluate accuracy on $\mathcal{D}_{test}$.
12:       Log training metrics (accuracy, memory usage, time).
13:    **end if**
14: **end for**

---

**Algorithm 4** FastOrthogonalize Procedures

---

1: **Procedure:** FASTORTHOGONALIZEINIT($\mathbf{W} \in \mathbb{R}^{d_1 \times d_2}$)
2: **Returns:** Orthogonalized weight matrix via QR decomposition
3: **if** $d_1 \geq d_2$ **then**
4:    $\mathbf{Q}, \mathbf{R} \leftarrow \text{QR}(\mathbf{W})$
5:    **return** $\mathbf{Q}[:,: d_2]$
6: **else**
7:    $\mathbf{Q}, \mathbf{R} \leftarrow \text{QR}(\mathbf{W}^T)$
8:    **return** $\mathbf{Q}[: d_1, :]^T$
9: **end if**
10:
11: **Procedure:** LIGHTWEIGHTNEWTONSCHULZ($\mathbf{X} \in \mathbb{R}^{d_1 \times d_2}$, steps $= 1$)
12: **Returns:** Orthogonalized matrix via Newton-Schulz iteration Eq. (15)
13: Normalize: $\mathbf{X} \leftarrow \mathbf{X}/\|\mathbf{X}\|$ if $\|\mathbf{X}\| > 1$
14: **for** $t = 1$ to steps **do**
15:    **if** $d_1 \geq d_2$ **then**
16:       $\mathbf{X} \leftarrow \mathbf{X}(1.5\mathbf{I} - 0.5\mathbf{X}^T\mathbf{X})$
17:    **else**
18:       $\mathbf{X} \leftarrow (1.5\mathbf{I} - 0.5\mathbf{X}\mathbf{X}^T)\mathbf{X}$
19:    **end if**
20: **end for**
21: **return** $\mathbf{X}$

---

## C   EXPERIMENTS

### C.1   HYPERPARAMETERS

Table 4: Hyperparameters of the Prospective Learning framework.

| Component | Parameter | Value |
|---|---|---|
| *Common Training Parameters* | | |
| Training Setup | Batch Size | 512 |
| | Training Epochs | 200 |
| | Learning Rate ($\alpha$) | 0.01 |
| | Architecture | [512, 128] |
| | Random Seed | 42 |
| *Prospective Learning Algorithm* | | |
| Algebraic Optimization | Regularization Parameter ($\lambda$) | $1 \times 10^{-4}$ |
| | Label Smoothing ($\epsilon$) | 0.01 |
| | Momentum Coefficient | 0.9 |
| Sparse Connectivity Initialization | Correlation Threshold ($\tau$) | 0.3 |
| | Small-World Rewiring Probability ($p$) | 0.1 |
| Adaptive Metaplasticity | Decay Factor ($\beta$) | 0.7 |
| | Sensitivity Parameter ($\gamma$) | 0.15 |
| | Inactivity Threshold ($\tau_{\text{inactive}}$) | $1 \times 10^{-3}$ |
| | Boosting Factor ($\xi$) | 3.0 |
| Stability Mechanisms | Orthogonalization Frequency | 5 epochs |
| | Orthogonalization Strength | 0.1 |
| | Spectral Normalization Max Norm | 2.0 |
| | ELU Alpha | 1.0 |
| | Activation Clipping Range | [-10, 10] |

Our hyperparameter selection follows three key principles: numerical stability, biological plausibility, and computational efficiency. The regularization parameter ensures matrix invertibility in the least-squares optimization while remaining small enough to preserve the gradient-free property. Label smoothing models biological uncertainty in neural target representations, following established practices in deep learning. For brain-inspired initialization, the correlation threshold balances network sparsity with representational capacity by connecting neurons based on input feature correlations, while small-world rewiring probability creates the characteristic high-clustering, short-path-length topology observed in cortical networks. The metaplasticity parameters implement synaptic memory principles where the decay factor models biological timescales of synaptic change, the sensitivity parameter prevents both over-stabilization and instability of frequently updated weights, and the boosting factor ensures inactive synapses maintain plasticity to prevent synaptic silencing. Stability mechanisms including orthogonalization frequency and spectral normalization provide gentle regularization without disrupting learned representations, while activation clipping prevents numerical overflow during matrix operations. All common training parameters follow standard practices to ensure fair comparison with baseline methods.

## C.2 EXTENDED EXPERIMENTAL ANALYSIS

Table 5: Performance comparison of different methods on MNIST, CIFAR-10, and CIFAR-100 datasets. M-Max denotes peak memory usage.

| Method | MNIST | | CIFAR-10 | | CIFAR-100 | |
|---|---|---|---|---|---|---|
| | Train-time(s) | M-Max(MB) | Train-time(s) | M-Max(MB) | Train-time(s) | M-Max(MB) |
| *Backpropagation-based* | | | | | | |
| SGD | **1.93±0.01** | 43.49±0.32 | **1.30±0.03** | 46.59±0.12 | **1.27±0.07** | 47.91±0.24 |
| Adam | 2.97±0.02 | 51.33±0.12 | 1.87±0.01 | 53.39±0.14 | 2.09±0.06 | 54.90±0.13 |
| AdamW | 3.69±0.13 | 51.95±0.34 | 2.07±0.04 | 54.14±0.27 | 2.43±0.07 | 54.76±0.18 |
| Muon | 4.51±0.11 | 48.14±0.17 | 3.41±0.09 | 50.14±0.11 | 3.26±0.03 | 51.40±0.27 |
| *Backpropagation alternatives* | | | | | | |
| ADMM | 7.87±0.04 | 54.70±0.01 | 4.77±0.07 | 56.11±0.13 | 4.96±0.06 | 57.95±0.11 |
| FF | 29.92±0.37 | 27.14±0.04 | 16.16±0.02 | 38.93±0.21 | 16.31±0.17 | 35.74±0.02 |
| NoProp | 16.04±0.15 | 160.07±2.77 | 11.22±0.09 | 159.85±2.11 | 12.52±0.09 | 161.63±1.48 |
| Ours | 5.05±0.04 | **20.77±0.10** | 2.84±0.04 | **30.70±0.07** | 2.86±0.06 | **32.38±0.14** |

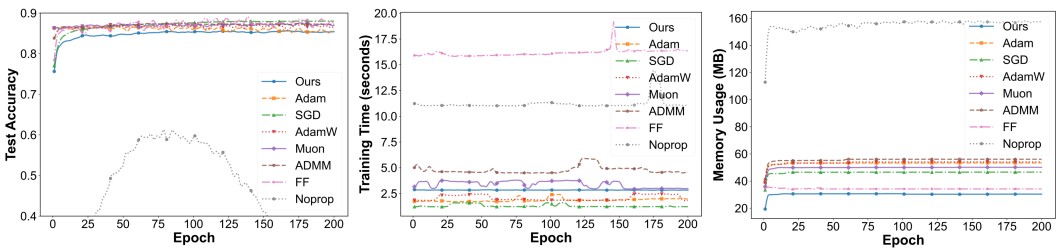

Figure 4: Training dynamics on CIFAR10 dataset showing (left) test accuracy convergence, (middle) per-epoch training time, and (right) memory usage throughout 200 epochs. All methods implemented using NumPy.

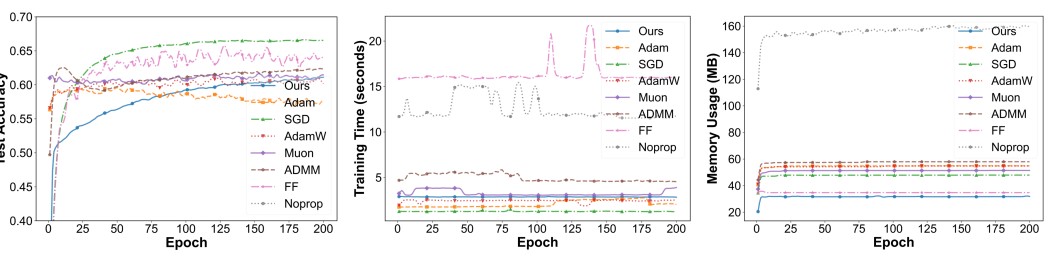

Figure 5: Training dynamics on CIFAR100 dataset showing (left) test accuracy convergence, (middle) per-epoch training time, and (right) memory usage throughout 200 epochs. All methods implemented using NumPy.

Table 5 provides additional performance metrics that further validate the efficiency advantages of our Prospective Learning framework. Our method consistently achieves the lowest peak memory consumption across all datasets with 20.77 MB, 30.70 MB, and 32.38 MB for MNIST, CIFAR-10, and CIFAR-100 respectively, representing substantial memory savings of 52.3%, 34.1%, and 32.4% compared to the most memory-efficient backpropagation baseline SGD. Regarding training time, our method demonstrates competitive performance compared to other gradient-based optimizers, achieving comparable or better efficiency than Muon on CIFAR-10 (2.84s vs 3.41s) and CIFAR-100 (2.86s vs 3.26s), while remaining competitive with Adam and AdamW across datasets. Most importantly, we significantly outperform other backpropagation alternatives, training 1.6× to 5.9×

faster than ADMM, 5.9× to 10.5× faster than Forward-Forward, and 3.2× to 5.7× faster than No-Prop. The modest computational overhead from our algebraic optimization is well justified by the dramatic memory reductions, making our approach highly suitable for resource-constrained deployment scenarios where memory limitations represent the primary bottleneck.

Figure 4 demonstrates the training dynamics of our Prospective Learning framework on the more challenging CIFAR-10 dataset. The convergence analysis reveals that our method achieves stable and competitive final accuracy comparable to traditional backpropagation methods, with smooth training dynamics free from the erratic behavior observed in NoProp. The per-epoch training time analysis shows our method maintains reasonable computational efficiency, consistently outperforming ADMM, Forward-Forward, and NoProp while remaining within acceptable bounds relative to gradient-based optimizers. Most significantly, the memory usage patterns confirm the sustained advantage of our framework throughout the entire training process. The consistent low memory footprint across all epochs, without the memory spikes characteristic of gradient-based methods, validates our approach's suitability for continuous deployment on memory-constrained devices.

Figure 5 extends our analysis to CIFAR-100, the most challenging dataset with 100 classes, further validating the robustness and scalability of our Prospective Learning framework under increased task complexity. The convergence analysis reveals that our method maintains stable training dynamics and achieves competitive final accuracy despite the significantly higher classification difficulty, demonstrating consistent performance across varying problem complexities. The per-epoch training time results show our approach continues to outperform backpropagation alternatives substantially, with particularly notable efficiency gains over Forward-Forward and ADMM, while remaining competitive with gradient-based methods. The consistent low memory footprint of our method, without the fluctuations observed in competing approaches, demonstrates exceptional stability for continuous operation on resource-constrained hardware. These results on the most demanding benchmark confirm that our memory efficiency advantages not only persist but become increasingly valuable as task complexity scales, establishing Prospective Learning as a viable solution for complex visual recognition tasks in memory-limited environments.

## C.3 HYPERPARAMETER SENSITIVITY ANALYSIS

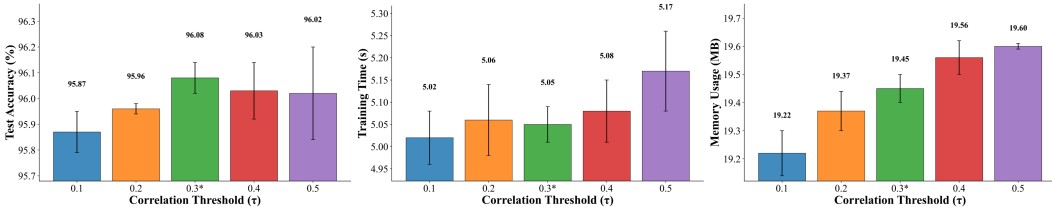

Figure 6: Ablation study of correlation threshold $\tau$ showing (left) test accuracy, (middle) training time, and (right) memory usage across different threshold values for sparse connectivity initialization.

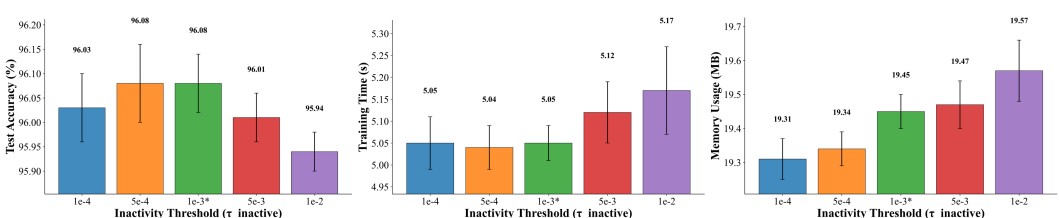

Figure 7: Analysis of metaplasticity inactivity threshold $\tau_{\text{inactive}}$ demonstrating (left) test accuracy, (middle) training time, and (right) memory usage across different threshold values for inactive weight boosting.

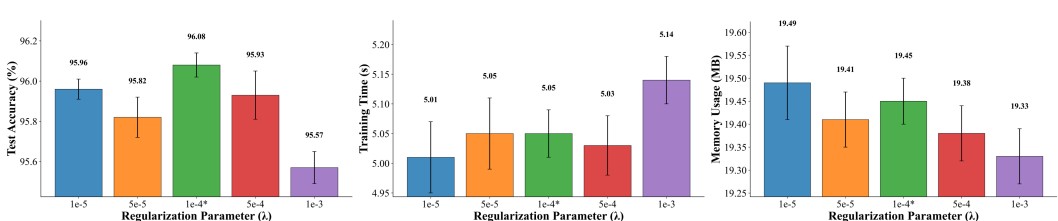

Figure 8: Regularization parameter $\lambda$ ablation revealing (left) test accuracy, (middle) training time, and (right) memory usage across different regularization strengths in the least-squares optimization.

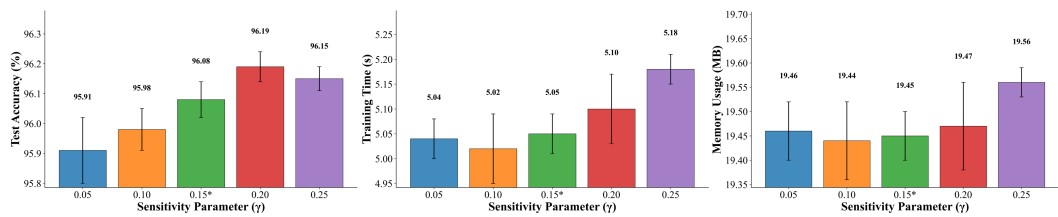

Figure 9: Sensitivity parameter $\gamma$ study showing (left) test accuracy, (middle) training time, and (right) memory usage across different metaplasticity modulation strengths.

**Correlation Threshold Analysis:** As shown in Figure 6, the correlation threshold $\tau$ controls the density of feature-based connections in sparse initialization by determining which input features are connected based on their correlation strength. According to the figure, $\tau = 0.3$ achieves optimal test accuracy with stable training times and memory usage, while extreme values either create overly sparse networks (low $\tau$) or approach dense connectivity that negates sparsity benefits (high $\tau$). This demonstrates that moderate correlation-based connectivity provides the best balance between network expressivity and computational efficiency.

**Inactivity Threshold Analysis:** As shown in Figure 7, the metaplasticity inactivity threshold $\tau_{\text{inactive}}$ determines when weights are considered inactive and receive learning rate boosting to prevent synaptic silencing. According to the figure, performance remains robust across 1e-4 to 1e-3 with consistent memory and timing, but degrades significantly at 1e-2 where excessive boosting disrupts training stability. This validates that the metaplasticity mechanism requires physiologically reasonable thresholds to maintain the stability-plasticity balance.

**Regularization Parameter Analysis:** As shown in Figure 8, the regularization parameter $\lambda$ ensures numerical stability in least-squares optimization by preventing singular matrix inversions during weight updates. According to the figure, $\lambda = 1e - 4$ provides optimal performance with stable computational costs, while values above 1e-4 show declining accuracy due to over-regularization constraining the optimization. This indicates that careful hyperparameter tuning is essential for maintaining the method's effectiveness.

**Sensitivity Parameter Analysis:** As shown in Figure 9, the metaplasticity sensitivity parameter $\gamma$ modulates the strength of adaptive learning rate adjustments based on individual weight activity history. According to the figure, performance peaks at $\gamma = 0.20$ rather than the default 0.15, showing nonmonotonic behavior with consistent computational overhead across all values. This reveals that optimal metaplasticity requires balanced modulation strength, with both insufficient and excessive sensitivity degrading learning outcomes.

Table 6: Scalability analysis of Prospective Learning across different MLP architectures on MNIST dataset.

| Architecture | Training Time (s) | Memory Usage (MB) |
| --- | --- | --- |
| [32, 32] | 1.5 | 10.0 |
| [64, 32] | 1.6 | 11.0 |
| [64, 64] | 1.6 | 12.5 |
| [128, 64] | 1.9 | 19.0 |
| [128, 128] | 2.0 | 20.0 |
| [256, 128] | 3.2 | 22.3 |
| [256, 256] | 3.7 | 27.1 |
| [512, 32] | 4.2 | 16.8 |
| [512, 64] | 4.5 | 17.1 |
| [512, 128] | 5.0 | 19.4 |

As shown in Table 6, the architecture scalability analysis reveals that our Prospective Learning framework demonstrates predictable computational scaling patterns across diverse network configurations. Training time exhibits strong correlation with input layer width, as evidenced by the consistent 5.0+ second training times for all 512-dimensional input architectures regardless of subsequent layer sizes, while smaller input dimensions maintain proportionally lower computational costs. Memory usage shows more complex scaling behavior that appears influenced by both total parameter count and architectural depth, with symmetric architectures like [256, 256] requiring higher memory (27.1 MB) compared to asymmetric variants like [512, 32] (16.8 MB) despite similar parameter counts. The framework maintains computational efficiency across all tested architectures, with even the largest configuration [512, 128] requiring only 19.4 MB memory and 5.0 seconds per epoch, demonstrating robust scalability for practical deployment scenarios where architectural flexibility is essential.

## D  LIMITATIONS AND FUTURE WORK

Our prospective learning framework is specifically designed for fully connected layer optimization, leveraging the mathematical elegance of direct algebraic solving for matrix-structured parameters. Like other specialized optimizers such as Muon (Liu et al., 2025) which requires hybrid usage with AdamW for embedding and output layers, our method targets a specific architectural component where its theoretical advantages are most pronounced. The $\mathcal{O}(d^3)$ complexity of algebraic optimization per layer becomes computationally demanding for networks with very wide hidden layers, potentially limiting scalability to extremely large fully-connected architectures. However, this limitation aligns well with our target deployment scenario of resource-constrained edge devices, where networks are typically smaller and the memory efficiency gains are most critical. The restriction to MLP layers represents a conscious design choice that enables the mathematical elegance of direct least-squares optimization while addressing the specific needs of edge computing applications.

While our method shows minor accuracy trade-offs compared to the best-performing baseline on each dataset, it maintains competitive performance relative to widely-used optimizers like Adam and Muon across all benchmarks. Our use of pretrained feature extractors for CIFAR experiments reflects two practical considerations: the current scope limitation to fully-connected layers, and the realistic edge deployment scenario where devices perform continual learning on top of existing pretrained representations. This experimental design mirrors real-world edge applications where devices must adapt learned features locally while maintaining computational efficiency.

Future research should focus on three key directions to enhance the framework's applicability and performance. First, developing techniques to improve accuracy while maintaining memory efficiency, potentially through advanced regularization strategies or hybrid optimization schemes. Second, optimizing the computational implementation of least-squares solving to reduce training time, possibly through parallel matrix operations or approximate solving methods. Third, extending the algebraic optimization principles beyond MLP layers to attention mechanism, which would enable broader application to modern deep learning architectures while preserving the memory advantages that make our brain-inspired approach valuable for resource-limited deployment scenarios.

## E  Biological Inspiration and Algorithmic Design

Our framework draws computational principles from neuroscience while prioritizing engineering practicality over strict biological fidelity. The core inspiration derives from prospective configuration mechanisms observed in biological neural networks, where learning occurs through a two-stage process: neurons first infer target activity patterns based on desired outcomes, then synaptic weights adjust to achieve these configurations. This contrasts with error-driven gradient descent, which iteratively adjusts weights based on loss derivatives. We translate this biological insight into our algebraic optimization framework by reformulating weight updates as direct target-solving problems Eq. (6)-Eq. (7), where each layer computes optimal weights to minimize the discrepancy between current activations and target configurations. This computational transformation eliminates the need for gradient computation and storage while preserving the essential credit assignment structure, demonstrating how biological principles can inform algorithmic innovation without requiring neuroscientific exactness.

The sparse connectivity initialization and adaptive metaplasticity mechanisms similarly reflect biological inspiration adapted for computational efficiency. Cortical networks exhibit structured sparsity with correlation-based local connections and small-world topology characterized by high clustering and short path lengths. We approximate these principles through our initialization strategy Eq. (9)-Eq. (15), which establishes connections based on input feature correlations and implements Watts-Strogatz rewiring, reducing parameters while maintaining representational capacity. Synaptic metaplasticity, wherein synaptic modification thresholds depend on prior activity history, inspires our adaptive learning rate modulation Eq. (16)-Eq. (19) that stabilizes frequently updated weights while maintaining plasticity in inactive synapses. These mechanisms are computational abstractions rather than faithful neural simulations, designed to leverage insights from biological learning systems to achieve practical benefits in memory-constrained environments. Our brain-inspired positioning thus reflects a pragmatic engineering philosophy: borrowing organizational principles from neuroscience to guide algorithm design, while maintaining focus on measurable performance improvements for edge computing applications.

## F  Extension to Convolutional Neural Networks

To validate the applicability of Prospective Learning outside of fully connected architectures, we demonstrate its extension to convolutional neural network training. Our approach leverages depthwise separable convolutions, which decompose standard convolutions into computationally efficient operations that can be algebraically optimized, providing a practical approach for memory-efficient CNN training on resource constrained edge devices.

### F.1  Mathematical Formulation for Convolutional Layers

The MobileNet architecture as a standard model for efficiently deploying CNNs on edge devices, greatly reducing computational requirements and achieving competitive accuracy through depthwise separable convolutions. Depthwise separable convolution decomposes standard convolution into two sequential operations: deep convolution and pointwise convolution, enabling our algebraic optimization framework to be applied to each component independently.

For depthwise convolution, each input channel $c$ is convolved independently with its own $3 \times 3$ kernel. Given input patches $\mathbf{P}^{(c)} \in \mathbb{R}^{n \times 9}$ extracted via im2col operation for channel $c$, where $n = B \cdot H_{out} \cdot W_{out}$ represents the batch size multiplied by output spatial dimensions, the convolution can be expressed as:

$$\mathbf{z}^{(c)} = \mathbf{P}^{(c)} \mathbf{w}^{(c)} + b^{(c)} \tag{38}$$

where $\mathbf{w}^{(c)} \in \mathbb{R}^9$ represents the flattened $3 \times 3$ kernel weights for channel $c$, and $b^{(c)}$ is the bias term. To apply Prospective Learning, we formulate the weight update as a regularized least-squares problem for each channel:

$$\Delta \boldsymbol{\theta}^{(c)} = \arg\min_{\Delta \boldsymbol{\theta}} \left\| \mathbf{X}_{aug}^{(c)} \Delta \boldsymbol{\theta} - (-\alpha \mathbf{e}^{(c)}) \right\|_2^2 + \lambda \|\Delta \boldsymbol{\theta}\|_2^2 \tag{39}$$

where $\mathbf{X}_{aug}^{(c)} = [\mathbf{P}^{(c)}, \mathbf{1}_n] \in \mathbb{R}^{n \times 10}$ augments the patch matrix with a bias column, $\mathbf{e}^{(c)} \in \mathbb{R}^n$ contains the backpropagated error signals for channel $c$, $\alpha$ is the learning rate, and $\lambda$ is the regularization parameter. This yields the closed-form solution:

$$\Delta\boldsymbol{\theta}^{(c)} = -\alpha \left( (\mathbf{X}_{aug}^{(c)})^T \mathbf{X}_{aug}^{(c)} + \lambda \mathbf{I} \right)^{-1} (\mathbf{X}_{aug}^{(c)})^T \mathbf{e}^{(c)} \tag{40}$$

where $\Delta\boldsymbol{\theta}^{(c)} = [\Delta\mathbf{w}^{(c)}; \Delta b^{(c)}] \in \mathbb{R}^{10}$ contains both kernel and bias updates. Critically, this formulation transforms a 4D convolutional operation into $C$ independent $10 \times 10$ linear systems, where $C$ is the number of input channels. The small matrix size enables efficient computation through Cholesky decomposition with caching, avoiding the cubic complexity bottleneck associated with large-scale matrix inversions. For $C$ channels, the computational cost is $O(C \cdot 10^3) = O(C)$, which is tractable for modest channel counts typical in edge deployment scenarios.

For pointwise ($1 \times 1$) convolution, the operation is mathematically equivalent to a fully-connected layer applied spatially. Given input activations $\mathbf{A} \in \mathbb{R}^{B \times H \times W \times C_{in}}$ reshaped to $\mathbf{X}_{flat} \in \mathbb{R}^{n \times C_{in}}$ where $n = B \cdot H \cdot W$, the pointwise convolution computes:

$$\mathbf{Z} = \mathbf{X}_{flat} \mathbf{W}_{pw} + \mathbf{b}_{pw} \tag{41}$$

where $\mathbf{W}_{pw} \in \mathbb{R}^{C_{in} \times C_{out}}$ and $\mathbf{b}_{pw} \in \mathbb{R}^{C_{out}}$. This directly applies our existing fully-connected layer optimization from Eq. (6)-Eq. (7) in the main text:

$$\Delta\boldsymbol{\Theta}_{pw} = -\alpha \left( \mathbf{X}_{aug}^T \mathbf{X}_{aug} + \lambda \mathbf{I} \right)^{-1} \mathbf{X}_{aug}^T \mathbf{E} \tag{42}$$

where $\mathbf{X}_{aug} = [\mathbf{X}_{flat}, \mathbf{1}_n] \in \mathbb{R}^{n \times (C_{in}+1)}$ includes the bias term, and $\mathbf{E} \in \mathbb{R}^{n \times C_{out}}$ contains the spatial error signals. The weight and bias updates are extracted as $\Delta\mathbf{W}_{pw} = \Delta\boldsymbol{\Theta}_{pw}[1 : C_{in}, :]$ and $\Delta\mathbf{b}_{pw} = \Delta\boldsymbol{\Theta}_{pw}[C_{in} + 1, :]$.

The key computational advantage of this factorization becomes apparent when comparing system sizes. A standard $3 \times 3$ convolution with $C_{in}$ input and $C_{out}$ output channels requires solving a linear system with $9 \cdot C_{in} \cdot C_{out}$ parameters. Depthwise separable convolution instead solves: (1) $C_{in}$ independent $10 \times 10$ systems for depthwise convolution with total complexity $O(C_{in})$, and (2) one $(C_{in} + 1) \times C_{out}$ system for pointwise convolution with complexity $O(C_{in}^3)$ when $C_{in} \approx C_{out}$. For modest channel counts ($C_{in}, C_{out} \leq 64$) typical in lightweight edge networks, this dramatically reduces the cubic complexity barrier while enabling gradient-free algebraic optimization.

### F.2 EXPERIMENTAL VALIDATION

We validate this CNN extension on MNIST using a lightweight MobileNet-inspired architecture consisting of three depthwise separable convolutional blocks (32 channels each with stride 2 for spatial downsampling after the first block) followed by global average pooling and two fully-connected layers ([32, 32] hidden dimensions). The network contains approximately 14K parameters and is trained for 300 epochs with batch size 128 using pure NumPy implementation on standard CPU hardware to match our edge deployment focus.

Table 7: Prospective Learning performance on lightweight CNN architecture (MNIST).

| Method | Test Acc (%) | Train Time (s) | Memory (MB) | M-Max(MB) |
|---|---|---|---|---|
| Prospective Learning | 88.42±0.12 | 43.23±0.19 | 6.15±0.05 | 7.33±0.02 |

Table 7 shows that Prospective Learning successfully extends to convolutional architectures, achieving 88.42% test accuracy on MNIST with the lightweight CNN. Our method achieves 43.23 seconds training time per epoch using NumPy on the CPU, while gradient based standard methods like Adam require unlimited training time under the same conditions. This is due to the computational overhead of storing intermediate activations for backpropagation and iterative gradient computation, making CNN training almost unfeasible on edge devices without GPU acceleration. The combination of 6.15 MB memory usage and tractable computational cost establishes prospective learning as a viable solution for enabling CNN training directly on resource-constrained edge devices. This ability is crucial for edge AI applications that require model adaptation, continuous learning, or personalization on devices, where transferring data to cloud servers is impractical due to privacy, latency,

or bandwidth limitations. Our algebraic optimization approach eliminates gradient storage through direct least-squares solving and leverages Cholesky decomposition with caching for the small depthwise systems, transforming CNN training from a computationally prohibitive operation to an actual edge deployment reality. These results establish a practical foundation for extending Prospective Learning to broader architectural patterns beyond fully-connected layers. Future work should explore applications to other efficient model designs, enabling gradient-free neural network training across diverse edge computing scenarios.

# G    LLM USAGE DECLARATION

We acknowledge the use of large language models (LLMs) in the preparation of this manuscript, in accordance with ICLR 2026 policies requiring disclosure of any LLM usage. Specifically, we employed LLMs solely for language polishing and writing refinement purposes, including grammar correction, sentence structure improvement, and clarity enhancement of our original content. All research ideas, methodologies, experimental designs, results analysis, and scientific conclusions presented in this work were conceived and developed entirely by the human authors. We take full responsibility for all contents written under our names, including any LLM-enhanced text, and confirm that no LLM-generated content constitutes plagiarism or scientific misconduct. The LLMs used served exclusively as writing assistance tools and were not involved in any aspect of the research process or intellectual contribution.

