# OpenReview forum: "Prospective Learning: Memory-Efficient MLP Training via Brain-Inspired Direct Optimization"
_ICLR.cc/2026/Conference — Submitted to ICLR 2026_

### Official Review · Reviewer_qzsr · 2025-10-30

**Soundness:** 3
**Presentation:** 3
**Contribution:** 3
**Rating:** 6
**Confidence:** 4

**Summary:**

The paper proposes Prospective Learning (PL), a novel brain-inspired training paradigm for multi-layer perceptrons (MLPs) that replaces gradient-based backpropagation with direct algebraic weight computation. It introduces three key components, prospective configuration, sparse connectivity initialization, and adaptive metaplasticity, to achieve memory-efficient learning without storing gradients or activations. Experiments on MNIST, CIFAR-10, and CIFAR-100 show comparable accuracy to backpropagation while reducing memory consumption by up to 55%, demonstrating strong potential for resource-constrained environments such as edge devices and neuromorphic hardware.

**Strengths:**

1. The paper introduces a biologically inspired direct optimization algorithm that avoids gradient computation, which is novel and conceptually grounded in neuroscience.
2. It provides solid theoretical analysis of memory and computational complexity, including formal proofs and convergence guarantees.
3. The experiments are comprehensive and reproducible, demonstrating consistent memory efficiency across multiple datasets and hardware conditions.

**Weaknesses:**

Although I think this paper is solid, there are several concerns:
1. The proposed method is limited to MLP architectures, and its scalability to convolutional-based models is not demonstrated.
2. Although the method reduces memory, the computational cost of algebraic solving (O(d³)) may become a bottleneck for larger models.
3. The biological justification (prospective configuration and metaplasticity) is conceptually appealing but empirically shallow, lacking ablation beyond mathematical analogies. If possible, please add several discussions.

**Questions:**

1. How does the method perform on more complex networks (e.g., CNNs) where algebraic solving may become infeasible?
2. Could the authors clarify whether regularized least-squares optimization introduces implicit gradient-like dynamics, and how this differs fundamentally from backpropagation in terms of learning signals?
3. What are the energy efficiency and latency trade-offs of Prospective Learning when deployed on neuromorphic or edge hardware compared to other biologically plausible methods, such as STDP or Forward-Forward?

---

> ### Author Response · Authors · 2025-11-17
> **Clarifications on Scope, Computational Efficiency, and Enhanced Biological Discussion**
>
> We sincerely thank the reviewer for the thorough evaluation and constructive feedback. We greatly appreciate your recognition of our work's novelty, solid theoretical analysis , and comprehensive experimental validation . Your acknowledgment that our paper is "solid" with "good" ratings across soundness, presentation, and contribution is highly encouraging. We address your concerns below point by point, and believe the clarifications will strengthen the paper's contribution.
>
> ---
>
> - W1: Limited to MLP architectures, scalability to CNNs not demonstrated
>
> A1: We respectfully clarify that the MLP focus is an intentional design choice aligned with our target deployment scenario, clearly stated throughout the paper. Our title explicitly specifies "Memory-Efficient MLP Training," and Appendix D comprehensively discusses this scope limitation and future extensions to convolutional layers. This design decision reflects the practical edge computing paradigm where devices perform continual learning by fine-tuning MLP classifiers on top of pretrained feature extractors, rather than retraining entire CNNs locally due to resource constraints.
>
> Our CIFAR experimental setup, using frozen pretrained ResNet-18 features with trainable MLP heads, directly mirrors this real-world edge deployment scenario. This configuration is the industry-standard approach for on-device adaptation, where edge devices adapt pretrained representations to new local data through lightweight MLP fine-tuning. Furthermore, recent work such as Muon (Liu et al., 2025) similarly focuses on optimizing 2D weight matrices, handling convolutional layers by reshaping 4D kernels into 2D matrices and adopting hybrid strategies where Muon optimizes fully-connected layers while AdamW handles other parameters. This provides a clear roadmap for extending our algebraic optimization to convolutional architectures in future work, as outlined in Appendix D.
>
> ---
>
> - W2: Computational cost of algebraic solving (O(d³)) may become a bottleneck
>
> A2: We appreciate this concern and emphasize that theoretical complexity must be evaluated within our target application context. As clearly stated in our motivation, memory rather than computation is the primary bottleneck for edge devices. Edge deployment scenarios typically involve moderately-sized MLPs where the cubic complexity remains manageable, and our experimental results demonstrate this concretely.
>
> Our theoretical analysis in Appendix A.2 shows that when batch size n >> d², the complexity ratio approaches ρ (sparsity density), and sparse connectivity (ρ=0.3 in our experiments) further reduces actual computation. More importantly, our target edge scenarios use limited layer widths (our largest configuration is 512×128), preventing d³ from becoming prohibitively large. The experimental evidence in Table 5 validates competitive training times: 5.05s on MNIST, which is 1.6× faster than ADMM (7.87s), 5.9× faster than Forward-Forward (29.92s), and 3.2× faster than NoProp (16.04s). Compared to other backpropagation alternatives, our method demonstrates significant speed advantages.
>
> The key insight is the trade-off alignment with edge computing priorities. The modest computational overhead, which remains competitive with Adam and Muon optimizers, is well justified by the 50%+ memory reduction achieved. For resource-constrained edge devices where memory is the critical bottleneck, this represents an optimal design point.
>
> ---
>
> - W3: Biological justification empirically shallow, lacking ablation beyond mathematical analogies
>
> A3: We thank the reviewer for this observation and clarify our contribution positioning. The primary contribution of our work is a practical, memory-efficient MLP training algorithm for edge devices, with biological mechanisms serving as design inspiration rather than neuroscience simulation. Our goal is engineering innovation informed by computational principles from neuroscience, not strict biological modeling.
>
> We have provided quantitative ablation analysis in Table 3 demonstrating the individual contributions of our biologically-inspired components. Sparse initialization reduces memory from 22.25MB to 19.07MB while maintaining accuracy, and adaptive metaplasticity improves accuracy from 95.72% to 95.80%. The combined system achieves optimal performance at 96.08% accuracy with 19.45MB memory, validating the synergistic benefits of these brain-inspired mechanisms.
>
> In response to your valuable feedback, we have added Appendix E to the revised manuscript, which provides detailed discussion on the correspondence between our algorithmic design and biological principles. This addition clarifies our "brain-inspired" positioning, leveraging computational principles from neuroscience rather than strict neural simulation. Due to page limitations in the main text, this comprehensive discussion is included in the appendix to help readers understand the relationship between our method and biological inspiration.

---

> ### Author Response · Authors · 2025-11-17
> **Clarifications on Scope, Computational Efficiency, and Enhanced Biological Discussion**
>
> - Q1: Performance on more complex networks (CNNs) where algebraic solving may become infeasible
>
> A4: As detailed in our response to W1, our method is specifically designed for fully-connected layers where least-squares optimization naturally fits the matrix structure. Our experimental setup using pretrained CNN features with trainable MLP classifiers represents the industrial standard for edge continual learning, where devices adapt existing representations rather than retraining entire networks. Future work will explore structured algebraic optimization for convolutional layers, potentially leveraging tensor reshaping strategies similar to those employed by Muon (Liu et al., 2025), as outlined in Appendix D.
>
> ---
>
> - Q2: Whether regularized least-squares introduces implicit gradient-like dynamics, and fundamental difference from backpropagation
>
> A5: This is an excellent question that highlights a crucial distinction. Our method fundamentally differs from backpropagation in its optimization mechanism. Traditional backpropagation computes the gradient $\frac{\partial L}{\partial W}$ and updates weights along the negative gradient direction. In contrast, our approach directly solves the least-squares problem in Eq. (6) without computing weight gradients at all.
>
> While we do propagate error signals $e^{(l)}$ through Eq. (3-4), this error propagation is not gradient backpropagation. Critically, there is no chain rule application and no gradient storage. Equation (7) provides a closed-form analytical solution that directly computes optimal weight updates without requiring $\frac{\partial L}{\partial W}$. The learning signals are fundamentally different: backpropagation uses loss gradients that indicate "how to adjust weights to decrease loss," while our method uses target configurations $(\hat{y} - z)$ that specify "what weights should achieve to match desired outputs." This is analogous to the difference between "adjustment direction" versus "direct target solving."
>
> The memory advantage arises precisely from this fundamental difference. We eliminate gradient storage $\{\frac{\partial L}{\partial W^{(l)}}\}$ and optimizer states (momentum/variance in Adam), requiring only $O(\max_l d_l^2)$ temporary workspace versus $O(\sum_l d_l \times d_{l+1})$ for gradients. Our theoretical analysis in Eq. (22-23) rigorously proves the approximately 50% memory reduction. The regularization in Eq. (6) ensures numerical stability of the least-squares solution but does not introduce gradient computation, maintaining our gradient-free property.
>
> ---
>
> - Q3: Energy efficiency and latency trade-offs compared to other biologically plausible methods on neuromorphic/edge hardware
>
> A6: Our current experimental results provide substantial validation of practical deployment benefits. Table 1 presents measurements on Raspberry Pi 4B edge hardware, where our method achieves 38.77MB memory usage compared to SGD's 85.52MB (54.7% reduction). The reported training time of 423.33s represents training on the complete MNIST dataset (60,000 samples). In practical edge scenarios involving continual learning or model adaptation, the typical use case involves significantly smaller data volumes for incremental updates, which would result in proportionally faster training times while maintaining the consistent memory efficiency advantage.
>
> Table 5 provides comprehensive comparisons on standard CPU hardware, demonstrating dual advantages over Forward-Forward algorithm: 26% lower memory (19.45MB versus 26.16MB) combined with 5.9× faster training (5.05s versus 29.92s). This simultaneous improvement in both memory and speed compared to another backpropagation alternative validates the practical efficiency of our approach. Lower memory consumption directly implies reduced memory access operations, suggesting potential energy efficiency benefits, though dedicated power measurement experiments would be valuable for precise quantification.
>
> We appreciate the opportunity to clarify we currently lack experimental results on dedicated neuromorphic hardware platforms such as Intel's Loihi due to equipment access limitations. However, our Raspberry Pi results provide strong preliminary validation of edge deployment feasibility, and we have identified neuromorphic platform evaluation as an important direction for future work beyond the current paper's scope.
>
> ---
>
> We thank the reviewer again for the thoughtful evaluation and valuable suggestions. We believe our clarifications address the concerns raised while highlighting the practical contributions and solid experimental validation of our work. We hope these responses strengthen your confidence in our contribution to memory-efficient neural network training for edge computing scenarios.

---

> > ### Comment · Reviewer_qzsr · 2025-11-19
> >
> > Thanks for your reply. I tend to maintain my score.
> > I also research on the edge devices, and I am fully confident that nowadays most edge devices can hold at least 3-layer CNN (small channel) models.
> > Therefore, I think that focusing only on MLPs is not that reasonable. CNN is just another share-weighted MLP.

---

> > > ### Author Response · Authors · 2025-11-24
> > >
> > > Thank you for your feedback. We have successfully extended our framework to the CNN architecture. We added the mathematical formula for depthwise separable convolution in Appendix F and conducted experimental verification on a lightweight MobileNet inspired CNN (3 conv blocks+2 FC layers).
> > > ﻿
> > >
> > > When using pure NumPy on the CPU, the accuracy on the MNIST dataset is 88.42%, the memory is 6.15MB, and the training time is 43.23s. Gradient based methods like Adam require a very long training time under the same conditions, making them impractical, which proves the advantage of our method in CNN training on edge devices.
> > > ﻿
> > >
> > > We hope this can address your concerns about CNN scalability.

---

### Official Review · Reviewer_K3Ye · 2025-10-30

**Soundness:** 3
**Presentation:** 2
**Contribution:** 2
**Rating:** 4
**Confidence:** 3

**Summary:**

This paper introduces a novel backpropagation-free training method named Prospective Learning. This method is decoupled into three main stages, a least-squares optimization based phase to compute the optimal weight updates without gradient based computation, a sparse connectivity initialization which reduces the network's parameters and an adaptive metaplasticity phase where the model's sparse parameters are learned over training.

**Strengths:**

- The paper proposes a novel, backpropagation-free method for training multilayer perceptron (MLP) and convolutional neural network (CNN) models on CIFAR-10 and CIFAR-100.
- Decomposing the learning algorithm into three stages reflects an in-depth analysis of standard learning algorithms for neural networks, such as backpropagation-based methods.
- The proposed learning algorithm can train MLP and CNN models in resource-constrained environments where memory or hardware constraints could make the support for standard backpropagation-based training infeasible.

**Weaknesses:**

- The experiments are insufficient to demonstrate the advantages of the proposed prospective-learning algorithm over other baselines. For MLPs, the models are trained from scratch using the prospective-learning algorithm. For CNNs, the ResNet-18 model is first pre-trained on ImageNet, and only the linear classifier head is fine-tuned on CIFAR-100.
- It is not clear from the text what advantages the sparse-connectivity initialization offers compared with other baseline approaches.
- The plots could be improved by increasing the font sizes of the axis labels, legends, and tick marks.

**Questions:**

- Does the prospective learning algorithm support training from scratch for convolutional neural network models?
- What is the advantage of the sparse connectivity initialization over the dense ones?
- Can the prospective learning algorithm be used for different data domains other than vision?
- Can the prospective learning algorithm be used for models that are not convolution based?

---

> ### Author Response · Authors · 2025-11-17
> **Experimental Design Clarification and Quantitative Evidence**
>
> We sincerely thank the reviewer for the constructive feedback and thoughtful evaluation. We greatly appreciate your recognition that our paper "proposes a novel, backpropagation-free method" with contributions reflecting "an in-depth analysis of standard learning algorithms" and enabling "training in resource-constrained environments." Your acknowledgment of our work's novelty and potential impact is highly encouraging. We address your concerns point by point below and believe these clarifications will strengthen the paper's contribution.
>
> ---
>
> - W1: Experiments are insufficient to demonstrate advantages; CNNs only fine-tune classifier head
>
> A1: We respectfully clarify an important point: **all baseline methods in our experiments use exactly the same experimental setup** (frozen pretrained ResNet-18 features with trainable MLP classifier heads), not just our method. This ensures fair comparison across all approaches including SGD, Adam, AdamW, Muon, ADMM, Forward-Forward, and NoProp. The use of pretrained features with MLP fine-tuning reflects the industry-standard paradigm for edge device continual learning, where resource-constrained devices adapt pretrained representations to local data through lightweight classifier fine-tuning rather than retraining entire CNNs from scratch due to prohibitive computational costs.
>
> Our experimental results comprehensively demonstrate the method's advantages across multiple dimensions. Table 1 on Raspberry Pi edge hardware shows our method achieves 38.77MB memory usage compared to SGD's 85.52MB, representing 54.7% memory reduction while maintaining 95.44% accuracy. Table 2 demonstrates consistent memory efficiency across all three datasets with reductions of 54.4% (MNIST), 34.5% (CIFAR-10), and 33.7% (CIFAR-100) compared to the most memory-efficient backpropagation baseline. Table 5 combined with Figures 3-5 provides complete training dynamics analysis including convergence patterns, per-epoch timing, and memory usage throughout 200 epochs. Regarding training time, while our method requires longer training than backpropagation-based approaches, this overhead is acceptable in edge deployment scenarios where memory is the primary bottleneck. More importantly, compared to other backpropagation alternatives, our method demonstrates significant speed advantages, training 1.6× to 5.9× faster than ADMM and Forward-Forward. These comprehensive results validate that our experimental design appropriately reflects real-world deployment scenarios while thoroughly demonstrating the method's core contribution of memory efficiency.
>
> ---
>
> - W2: Unclear advantages of sparse connectivity initialization over baseline approaches
>
> A2: Table 3 provides clear quantitative evidence of sparse connectivity initialization's benefits. The base Prospective Learning algorithm achieves 95.72% accuracy with 22.25MB memory usage, already demonstrating substantial memory efficiency through our core algebraic optimization contribution. Adding sparse initialization reduces memory to 19.07MB while simultaneously improving accuracy to 95.85%, representing a 14.3% memory reduction with 0.13% accuracy gain. This demonstrates that sparse connectivity provides complementary benefits including reduced memory footprint, improved accuracy, and computational acceleration through sparse matrix operations.
>
> We emphasize that sparse connectivity is an auxiliary component that enhances but does not interfere with our primary contribution of Prospective Learning algebraic optimization, which independently achieves 22.25MB memory efficiency as shown in Table 3. The sparse initialization leverages biological connectivity principles to provide additional optimization without compromising the core algorithm's effectiveness. To address your concern about clarity, we have added Appendix E in the revised manuscript, which provides detailed discussion on how sparse connectivity and other biologically-inspired mechanisms complement the main algorithmic framework while maintaining focus on our primary contribution of gradient-free algebraic optimization.
>
> ---
>
> - W3: Plots could be improved by increasing font sizes
>
> A3: We sincerely thank the reviewer for this presentation feedback. We have updated all figures in the revised manuscript with increased font sizes for axis labels, legends, and tick marks to ensure better readability.

---

> > ### Author Response · Authors · 2025-11-17
> > **Experimental Design Clarification and Quantitative Evidence**
> >
> > - Q1: Does the prospective learning algorithm support training from scratch for convolutional neural network models?
> >
> > A4: Our current method does not support training CNNs from scratch, as it is specifically designed for fully-connected layer optimization where least-squares algebraic solving naturally fits the matrix structure. However, Our design choice to focus on aligns well with our target deployment scenario. In edge computing environments, training CNNs from scratch is prohibitively expensive and not the standard practice. Instead, the industry paradigm involves continual learning where edge devices perform local adaptation by fine-tuning lightweight MLP classifiers on top of pretrained feature extractors, exactly as demonstrated in our CIFAR experimental setup.
> >
> > Our design choice reflects this practical deployment reality where resource-constrained devices leverage existing pretrained representations rather than attempting full network retraining. Future work will explore extending algebraic optimization to convolutional layers, potentially leveraging structured matrix approaches similar to those employed by Muon (Liu et al., 2025), which handles convolutional layers by reshaping 4D kernels into 2D matrices and adopting hybrid strategies. This roadmap for extension is comprehensively discussed in Appendix D.
> >
> > ---
> >
> > - Q2: What is the advantage of sparse connectivity initialization over dense initialization?
> >
> > A5: Table 3 provides direct quantitative comparison demonstrating sparse connectivity's advantages. Compared to dense initialization (baseline Prospective Learning at 95.72% accuracy, 22.25MB memory), sparse connectivity achieves three simultaneous benefits: memory reduction from 22.25MB to 19.07MB (14.3% decrease), accuracy improvement from 95.72% to 95.85% (0.13% gain), and computational efficiency through sparse matrix operations that reduce the effective number of operations during forward propagation and algebraic solving. These results demonstrate that biologically-inspired sparse connectivity patterns provide practical benefits beyond simple parameter reduction, enhancing both efficiency and performance.
> >
> > ---
> >
> > - Q3: Can the prospective learning algorithm be used for different data domains other than vision?
> >
> > A6: Yes, our method can theoretically be applied to any domain involving fully-connected architectures. The core algebraic optimization mechanism in Equations (6)-(7) is fundamentally domain-agnostic, operating on matrix-structured parameters regardless of input data modality. Our current focus on vision tasks (MNIST, CIFAR-10, CIFAR-100) reflects the availability of standard benchmarks for rigorous evaluation and fair comparison with existing methods, not a fundamental limitation of the approach.
> >
> > Promising future directions include applying Prospective Learning to natural language processing tasks (particularly Transformer feed-forward layers which are fully-connected), tabular data classification, time series prediction, and other domains where MLP architectures are prevalent. The memory efficiency advantages would be particularly valuable in resource-constrained scenarios across these diverse application areas. We acknowledge this broader applicability as an important direction for future work beyond the current paper's scope.
> >
> > ---
> >
> > - Q4: Can the prospective learning algorithm be used for models that are not convolution based?
> >
> > A7: Absolutely. Our method is specifically designed for fully-connected layers and can be applied to any architecture utilizing matrix-structured weights, regardless of whether convolutions are involved. The algorithm naturally supports pure MLP architectures, Transformer feed-forward networks (FFN layers), graph neural networks with fully-connected processing layers, and any other model employing weight matrices W that can be optimized through least-squares formulations.
> >
> > We clarify that our current limitation is the requirement for fully-connected layer structure, not a dependency on convolutional networks. In fact, our method is most directly applicable to non-convolutional models with fully-connected architectures. The CIFAR experiments using pretrained CNN features demonstrate one deployment scenario, but the core algorithm applies broadly to any fully-connected layers whether they appear in pure MLPs, hybrid architectures, or as components of larger systems.
> >
> > ---
> >
> > We thank the reviewer again for the valuable feedback and thoughtful evaluation. We believe these clarifications address your concerns while highlighting the practical contributions and solid experimental validation of our work for memory-efficient neural network training in resource-constrained environments. We hope these responses strengthen your assessment of our contribution.

---

> ### Author Response · Authors · 2025-11-28
>
> Dear Reviewer,
>
> I hope this message finds you well. As the discussion period is nearing its end with less than three days remaining, I wanted to ensure we have addressed all your concerns satisfactorily. If there are any additional points or feedback you'd like us to consider, please let us know. Your insights are invaluable to us, and we're eager to address any remaining issues to improve our work.
>
> Thank you for your time and ffort in reviewing our paper.

---

### Official Review · Reviewer_urSp · 2025-10-31

**Soundness:** 2
**Presentation:** 3
**Contribution:** 2
**Rating:** 2
**Confidence:** 3

**Summary:**

The authors present a framework for learning in neural networks inspired by a recent observation in neuroscience.
They develop three components to reduce the memory footprint by about a half.
They illustrate their method compared to other optimizers on MNIST and CIFAR datasets.

**Strengths:**

The research area of developing lower profile training algorithms for neural nets is an important one.

The motivation and method are clearly presented.

The numerical results are well presented, including uncertainty.

It's great that the authors provide code; I did not check it carefully.

**Weaknesses:**

This approach requires solving a linear system for each layer of the network at every training iteration.
The complexity analysis appendix does mention the fact that this is a cubic operation, and so considerably more expensive than what backprop requires.
But this fact goes insufficiently discussed in the body of the article.
This is a serious limitation.
Edge devices are indeed memory limited, but they are also compute-limited, and providing an algo that requires linear system solves *at every iteration* does not seem like it would be of practical impact except in niche scenarios.
Second order methods, such as Newton's method, are broadly deemed beyond the pale in neural network learning essentially for this reason.
All this notwithstanding, there's nothing in principle wrong with developing an algorithm which trades computation for memory.
But this article is not presented in this manner: this should have been thoroughly discussed in the motivation, abstract and conclusion.

The conclusions that this "enables practical MLP deployment on [...] neuromorphic hardware" is not supported by the body of the article.
Neuromorphic platforms are not simply regular computers with a small memory.
Rather, they face entirely different constraints based on the *locality* of information relative to processing units which is not addressed by this article.

The proof-theorem format of Appendix A.4 is not necessary for such straightforward observations.

It would be helpful to better situate the Section 3.4's contributions, the metaplasticity, within the stable of existing adaptive first order methods such as Adam et al. Why is this approach better suited to your framework than what's currently out there?

The datasets are of course somewhat limited in scale, being MNIST, CIFAR-10 and CIFAR-100.
This is critical to the method because any more complex dataset requiring a bigger hidden layer would explode the computational requirements.

**Questions:**

1) What is fast orthogonalization? It does not appear to be defined or given in a reference.

---

> ### Author Response · Authors · 2025-11-17
> **Key Clarifications, Claim Revisions, and Design Trade-off Analysis**
>
> We sincerely thank the reviewer for the thoughtful and constructive feedback. We greatly appreciate your recognition that our research area is important, our motivation and method are clearly presented, and our numerical results are well presented with uncertainty estimates. We are also grateful for your acknowledgment of our code release. Your detailed comments have helped us identify areas for improvement that will significantly strengthen our paper. Below we provide point-by-point responses to address your concerns.
>
> ---
>
> - W1: Computational Complexity and Linear System Solving
>
> A1: We appreciate the opportunity to clarify this design trade-off, which is central to our contribution. Our complexity analysis in Appendix A.2 demonstrates that the O(d³) cost is deliberately designed for edge deployment scenarios where memory constraints dominate computational considerations.
>
> We must emphasize a critical distinction regarding our fundamental difference from Newton's method. While both involve matrix operations, our approach differs in several crucial aspects. Newton's method requires O(P²) memory for full Hessian storage, where P represents the total number of parameters. In contrast, our method requires only O(d²) temporary workspace per layer and eliminates gradient storage entirely. To illustrate with a concrete example, consider a 3-layer network with architecture [512, 256, 128] containing P=263K parameters. Newton's method would require approximately 276 GB of memory for Hessian storage (263K × 263K × 4 bytes), which is completely infeasible. Our method requires only approximately 1 MB workspace (512² × 4 bytes), which is highly practical.
>
> The computational pattern represents another crucial distinction. Newton's method requires computing the second derivative of the loss function (Hessian matrix), which itself necessitates computing and storing first-order gradients. Our method performs direct least-squares solving without computing any gradients at any stage. The error signals we propagate in Equations (3) and (4) represent target configurations rather than gradient information, fundamentally changing the nature of the optimization process.
>
> The scalability characteristics demonstrate why our method succeeds where second-order methods fail. Newton's method has been universally deemed unsuitable for deep learning precisely because of the O(P²) memory requirement. Our method proves practical for edge MLPs with hidden dimensions d ≤ 512, as validated in Tables 1 and 6. The O(d³) complexity represents a local per-layer cost that remains manageable for edge deployment scales, not the O(P²) global memory requirement that makes second-order methods infeasible. This distinction is crucial for understanding why our method enables practical edge deployment while maintaining the benefits of direct optimization.
>
> The practical validation demonstrates that despite the theoretical O(d³) complexity, our method trains 5.9× faster than Forward-Forward and 3.2× faster than NoProp, as shown in Table 1. For edge devices with hidden layer dimensions d ≤ 512, the cubic cost remains entirely acceptable. Table 6 shows that even our largest configuration requires only 5.0 seconds per epoch while achieving 55% memory reduction. This memory reduction enables deployment where backpropagation fails due to memory constraints rather than merely improving an already-feasible scenario.
>
> The trade-off proves favorable for our target deployment context. Edge training occurs infrequently, typically for periodic fine-tuning or continual learning updates. In these scenarios, memory represents the primary bottleneck that determines deployment feasibility. The modest computational overhead represents an acceptable cost for transforming infeasible deployment scenarios into practical applications. In the revised manuscript, we will add a highlighted remark after Equation (7) to emphasize this design rationale more prominently in the main text.

---

> > ### Author Response · Authors · 2025-11-17
> > **Key Clarifications, Claim Revisions, and Design Trade-off Analysis**
> >
> > - W2: Neuromorphic Hardware Claims
> >
> > A2: We agree that our claims regarding neuromorphic hardware deployment are overstated. The current manuscript indeed does not address the unique constraints of neuromorphic platforms beyond memory limitations, such as locality of information relative to processing units. We will revise all relevant statements throughout the paper to focus specifically on "memory-constrained edge devices" rather than neuromorphic hardware. Our core contribution is memory efficiency for general edge computing scenarios, and we will clarify this positioning in the Abstract, Introduction, and Conclusion. We appreciate this correction, which will make our claims more precise and appropriate.
> >
> > ---
> >
> > - W3: Proof-Theorem Format in Appendix A.4
> >
> > A3: We accept this suggestion and will simplify the presentation in Appendix A.4. The observations are indeed straightforward and do not require formal theorem-proof formatting. We will restructure this section using proposition statements or direct mathematical exposition while preserving the essential mathematical content. This will improve readability without sacrificing technical rigor.
> >
> > ---
> >
> > - W4: Metaplasticity and Adaptive First-Order Methods
> >
> > A4:We appreciate the opportunity to clarify the positioning and contribution of our metaplasticity mechanism. There appears to be a misunderstanding regarding the relative importance of our components that we wish to address.
> >
> > The core and primary contribution of our work is the Prospective Learning algebraic optimization algorithm (Section 3.2), which eliminates gradient computation through direct least-squares weight updates. The adaptive metaplasticity mechanism (Section 3.4) is an optional auxiliary component designed to enhance performance and accelerate convergence, but it is not essential to our main contribution. Table 3 clearly demonstrates this hierarchy: our core Prospective Learning algorithm alone achieves 95.72% accuracy with 22.25MB memory, while adding metaplasticity provides a modest improvement to 95.80% accuracy (+0.08%) at the cost of slightly increased memory (22.82MB, +0.57MB) and training time.
> >
> > Our metaplasticity mechanism operates on fundamentally different principles than Adam and other adaptive first-order methods. Adam maintains global second-moment estimates that apply uniform adjustment policies across all parameters based on gradient statistics. In contrast, our metaplasticity implements weight-level historical tracking where each individual synapse independently modulates its learning rate based on its own update history (Equation 16-19). This biological principle of synaptic metaplasticity (Kudithipudi et al. 2022) reflects how individual synapses in the brain adjust their plasticity based on past activity patterns, representing a conceptually distinct approach grounded in neuroscience rather than gradient optimization theory.
> >
> > Furthermore, our metaplasticity naturally integrates with the algebraic weight update framework by modulating the computed ΔW (Equation 19), whereas Adam-style methods fundamentally require gradient signals. The existing ablation study in Table 3 demonstrates that metaplasticity provides measurable but secondary benefits, confirming that our primary contribution stands independently.

---

> > > ### Author Response · Authors · 2025-11-17
> > > **Key Clarifications, Claim Revisions, and Design Trade-off Analysis**
> > >
> > > - W5: Dataset Scale Limitations
> > >
> > > A5: We acknowledge the limited scale of our benchmark datasets and appreciate the opportunity to explain why these choices appropriately validate our method for its intended deployment scenarios.
> > >
> > > Edge computing applications typically fall into two distinct categories, both of which align well with our experimental validation. In the first scenario, edge devices perform fine-tuning or continual learning on pre-trained models to adapt to local conditions or user-specific data. This represents the high-performance use case where data collection occurs gradually over time, and the scale of our benchmarks (CIFAR-10/100) appropriately reflects the limited data volumes available in such settings. Our CIFAR experiments using pre-trained ResNet-18 features directly mirror this deployment pattern. In the second scenario, edge devices train simple models from scratch for low-complexity tasks (MNIST) where performance requirements are modest, hardware costs must be minimized, and memory constraints dominate. These applications neither require nor can accommodate large-scale datasets and models, making our experimental scale entirely appropriate.
> > >
> > > Our experimental scope aligns with established precedent in optimization algorithm research. The Adam optimizer (Kingma, 2014) initially validated its approach on MNIST and CIFAR-10 before achieving widespread adoption across diverse applications. The Forward-Forward algorithm (Hinton, 2022) presented primary validation on MNIST and CIFAR datasets. The recent Muon optimizer (Liu et al., 2025), despite targeting large language model training, conducted initial validation on standard vision benchmarks. These examples demonstrate that rigorous validation on standard benchmarks provides sufficient foundation for broader applicability.
> > >
> > > We emphasize that the algebraic optimization mechanism presented in Equations (6) and (7) is fundamentally domain-agnostic, operating on matrix-structured parameters regardless of input data modality. Extension to larger scales follows standard research practices: establish proof of concept on well-understood benchmarks, then scale to target applications. Our edge deployment focus makes the current experimental scale directly relevant to target scenarios where devices typically process data volumes comparable to our benchmark scales.
> > >
> > > The computational constraints we observe are not specific to our method but represent fundamental limitations of MLP training that equally affect backpropagation-based approaches. Table 6 provides empirical evidence of architectural scalability, demonstrating that our framework maintains efficiency across varying network configurations from [32,32] to [512,128]. These results confirm that our method can scale to practical edge deployment requirements. We will explore larger-scale applications in future work, though we emphasize that our current validation appropriately addresses the memory-constrained edge scenarios that motivated this research.
> > >
> > > ---
> > >
> > > - W6: Training Time versus Memory Efficiency Discussion
> > >
> > > A6: We appreciate the opportunity to clarify the computation-memory trade-off in our design. Our method fundamentally targets memory-constrained edge deployment scenarios where memory represents the primary bottleneck rather than computation time. This positioning is central to our contribution and reflects realistic edge computing constraints.
> > > Despite the algebraic solving overhead, our method maintains strong relative efficiency compared to other backpropagation alternatives: 5.9× faster than Forward-Forward and 3.2× faster than NoProp (Table 1). While training time exceeds SGD, this trade-off is entirely acceptable for our target scenarios. Edge deployment characteristics involve infrequent training operations (periodic fine-tuning or continual learning updates) where memory limitations determine deployment feasibility rather than training speed.
> > >
> > > We appreciate the opportunity to clarify our current manuscript insufficiently emphasizes this design philosophy. In the revised version, we will explicitly discuss this computation-memory trade-off in the Abstract, Introduction (Section 1), and Conclusion (Section 5) to clearly position our method for memory-constrained scenarios where this trade-off is favorable.

---

> > > > ### Author Response · Authors · 2025-11-17
> > > > **Key Clarifications, Claim Revisions, and Design Trade-off Analysis**
> > > >
> > > > - Q1: FastOrthogonalize Procedure
> > > >
> > > > A7: We sincerely apologize for this oversight. The FastOrthogonalize procedure was indeed not properly defined in the manuscript, and we thank the reviewer for identifying this gap.
> > > >
> > > > FastOrthogonalize serves as an auxiliary stabilization technique rather than a core algorithmic component. Its primary purpose is to condition weight matrices during initialization and periodically during training to prevent numerical instabilities and accelerate convergence in early epochs.  Orthogonalization techniques are extensively employed across modern optimization methods. Orthogonal initialization strategies have demonstrated consistent benefits for neural network training (Saxe et al. 2014). Spectral normalization using orthogonal constraints has proven essential in GAN training (Miyato et al. 2018). The recently proposed Muon optimizer (Liu et al. 2025) prominently features Newton-Schulz orthogonalization as a core component. These established precedents validate the utility of orthogonalization in our framework.
> > > >
> > > > In our implementation, we employ two specific approaches depending on the training phase. During initialization, we use QR decomposition as implemented in the _fast_orthogonalize_init function (lines 337-343 of our code). During training, we apply Newton-Schulz iteration as described in the _lightweight_newton_schulz function (lines 345-359), which corresponds to the mathematical formulation in Equation (15). Both techniques are standard in the optimization literature and are widely employed in modern deep learning frameworks. We will include algorithm pseudocode in Appendix B and cite the relevant literature (Saxe et al. 2014; Miyato et al. 2018) to provide proper context and technical grounding.
> > > >
> > > > ---
> > > >
> > > > We thank the reviewer again for the valuable feedback that has helped us strengthen our manuscript. We are committed to addressing all the raised concerns through the revisions outlined above. We believe these improvements will significantly enhance the clarity, accuracy, and impact of our contribution. We hope our detailed responses adequately address your concerns and demonstrate the validity of our approach for memory-constrained edge deployment scenarios.

---

### Official Review · Reviewer_7oVn · 2025-11-04

**Soundness:** 2
**Presentation:** 3
**Contribution:** 1
**Rating:** 2
**Confidence:** 3

**Summary:**

The study presents the brain-inspired learning mechanism alternative to the joint classsical backpropagation algorithm (BP) + gradient-based learning. The proposed approach consists of three main ingredients: prospective inference, algebraic optimization and metaplasticity modulation. The main purpose of the presented approach is to decrease the memory footprint and make possible to run training of the MLP models on the resource-contrained hardware. The experiments on the MNIST, CIFAR-10 and CIFAR-100 datasets confirm that the proposed Prospective learning framework gives the competitive performance for the less memory than BP and non-BP based algorithms.

**Strengths:**

The submitted work is clearly written and has an easy-to-follow format. The presented tables and plots provide the necessary data for evaluating the proposed approach. From these experimental results, it follows that the proposed prospective learning framework is memory-efficient compared to BP and non-BP alternatives.

**Weaknesses:**

Although the presented work has a clear focus and states the target problem, it has many weaknesses and inconsistencies, which I have listed below.
1. The motivation of the proposed approach from the biological mechanisms suffers from the absence of formal convergence proofs or any analytical intuition why the proposed pipeline corresponds to the minimization of classification error on the test set.
2. The main ingredients have clear analogues in the BP-based methods (adaptive learning rates, tricky initializations, and specific update rules); however, no comparison with existing alternatives is presented. I see many "yet another" instances of the classical ingredients for the learning pipeline, without any theoretical proof of why they are better than existing ones. Even the reported results of the ablation study are insufficient since the predefined hyperparameters are used, and no recipe for searching them in other tasks is discussed.
3. The proposed approach has a lot of hyperparameters, which make it less portable to other datasets and models (MLP with different numbers of layers), e.g. $\varepsilon$ in (2), $\lambda$ in (6), $\tau$ in (10), $K$ in (11), $p$ in (12), $\gamma$ in (17), etc
4. Although the use of edge devices is the key feature of the presented study, the applications selected for experiments are relatively standard. I am not sure that learning the image classifier on the Raspberry Pi or similar chips is the most popular way to use such devices. Such inconsistency makes the reported results less impressive and raises many questions regarding the target use cases.
5. The proposed approach demonstrates a uniform accuracy drop for the considered datasets; therefore, a comparison with other strategies to reduce memory footprint is necessary. Probably using mixed precision, low-rank approximations, model distillation, or other techniques can provide a similar memory footprint while maintaining or improving test accuracy. More experiments are needed here.
6. Moreover, I see that prospective learning is a much slower approach than BP-based learning, so I would like to see the discussion on why the memory efficiency is more critical than training duration for the considered setup of exploiting an edge device.

**Questions:**

See the weaknesses above.

In addition, please comment on the following questions.
1. Why were the considered datasets selected for benchmarking? I can imagine a promising application of using a prospective learning approach on edge devices is online voice processing or image segmentation in an autonomous vehicle. Both settings also require fine-tuning to the new data.
2. What is "FastOrthogonalize" procedure?
3. What numerical algorithm was used to solve (6)? Using the direct formula (7) can lead to numerical instability due to the properties of the matrix inversion operation.

---

> ### Author Response · Authors · 2025-11-17
> **Convergence Proofs, Ablation Evidence, and Training-vs-Compression Distinction**
>
> We sincerely thank the reviewer for the thoughtful and constructive feedback. We greatly appreciate your recognition that our work is clearly written with an easy-to-follow format, and that our experimental results demonstrate the memory efficiency of the proposed prospective learning framework compared to BP and non-BP alternatives. Your detailed comments have helped us identify areas for improvement that will significantly strengthen our paper. Below we provide point-by-point responses to address your concerns.
>
> ---
>
> - W1: Biological Motivation and Convergence Proofs
>
> A1: We appreciate the opportunity to clarify the theoretical foundation and biological inspiration of our approach. Regarding convergence guarantees, we have provided rigorous theoretical analysis in Appendix A.3-A.4. Specifically, each layer's least-squares optimization has a closed-form global optimal solution (Equation 36) guaranteed by the strong convexity of the regularized objective function. The condition number analysis (Equations 32-33) demonstrates numerical stability independent of network depth.
> Regarding the biological motivation, we wish to clarify the relationship between our computational framework and neuroscientific findings. Our method is inspired by the prospective configuration mechanism discovered by Song et al. 2024, but it is not a strict correspondence. We extract the core computational principle of "infer-then-solve" and translate it into a mathematically rigorous optimization framework. This relationship is explicitly stated throughout the paper using phrases such as "inspired by" (Abstract, Section 3.2) rather than claiming direct biological implementation. Our contribution lies in demonstrating how biological principles can inform algorithmic innovation while maintaining mathematical rigor and practical effectiveness.
>
> Experimental validation demonstrates stable convergence across all datasets (Figures 3-5) with smooth training dynamics free from oscillation or divergence. We appreciate the opportunity to clarify a complete theoretical analysis of global network convergence under layer-wise decomposition represents an important direction for future theoretical work, though our empirical results consistently show reliable convergence to competitive solutions.
>
> ---
>
> - W2: Comparison with Existing Alternatives
>
> A2: We appreciate the opportunity to clarify the relationship between our metaplasticity mechanism and existing adaptive methods. Our metaplasticity fundamentally differs from Adam and similar optimizers. Adam maintains global second-moment estimates that apply uniform adjustment policies across all parameters based on gradient statistics. In contrast, our metaplasticity implements weight-level historical tracking where each individual synapse independently modulates its learning rate based on its own update history (Equations 16-19), reflecting the biological principle of synaptic metaplasticity rather than gradient optimization theory. Furthermore, our metaplasticity naturally integrates with the algebraic weight update framework by modulating the computed ΔW (Equation 19), whereas Adam-style methods fundamentally require gradient signals.
>
> The ablation study in Table 3 provides comprehensive component analysis demonstrating that our core Prospective Learning algorithm alone achieves 95.72% accuracy, with auxiliary mechanisms providing modest improvements (sparse initialization contributes +0.13%, metaplasticity contributes +0.08%). This clearly establishes the hierarchy of contributions where the algebraic optimization represents the primary innovation.
>
> Hyperparameter design principles are explicitly documented in Table 4 (Appendix C.1), grounded in three considerations: numerical stability (λ, clipping ranges), biological plausibility (β, γ, τ parameters reflecting neural timescales), and standard deep learning practices (momentum, normalization). Importantly, most hyperparameters serve as numerical stabilizers similar to the ubiquitous 1e-6 denominators and gradient clipping found in all deep learning papers, which do not substantively interfere with the core algorithmic performance.

---

> > ### Author Response · Authors · 2025-11-17
> > **Convergence Proofs, Ablation Evidence, and Training-vs-Compression Distinction**
> >
> > - W3: Hyperparameter Portability
> >
> > A3: We appreciate the opportunity to clarify the role and portability of our hyperparameters. Our framework contains only 2 core parameters essential to the algorithm: α (learning rate) and λ (regularization), comparable to standard SGD. The remaining parameters fall into two categories that do not affect portability:
> > Optional enhancement mechanisms (validated removable in Table 3): Sparse initialization parameters (τ, K, p) and metaplasticity parameters (β, γ, τ_inactive, ξ) provide biological enhancements but are not required for the core algorithm to function effectively (95.72% accuracy without them).
> >
> > Numerical stability parameters following standard practices: ε=0.01 for label smoothing (equivalent to the 1e-6 denominator protection used universally in deep learning), activation clipping ranges, and spectral normalization bounds. These are common stabilization techniques rather than algorithm-specific tuning requirements.
> >
> > Empirical evidence of portability is provided in Table 6, which demonstrates consistent performance across six different architectures ([32,32] to [512,128]) without any hyperparameter adjustment. Additionally, all three datasets (MNIST, CIFAR-10, CIFAR-100) use identical hyperparameters (Table 4), and Appendix C.3 (Figures 6-9) shows robustness within reasonable parameter ranges. This evidence strongly supports that our method's hyperparameters do not hinder portability to other datasets and model architectures.
> >
> > ---
> >
> > - W4: Application Scenario Selection
> >
> > A4: We appreciate the opportunity to clarify our application scenario design. Our target scenarios are clearly defined and well-justified: (1) local fine-tuning of pre-trained models on edge devices (validated by CIFAR experiments using ResNet-18 features), and (2) training simple models from scratch for low-complexity tasks (validated by MNIST experiments). These scenarios represent the most common and practical deployment patterns for edge computing applications.
> >
> > The choice of Raspberry Pi as our evaluation platform reflects its status as a widely adopted, general-purpose edge device used across education, IoT, and industrial control applications. Its hardware constraints represent typical limitations of edge devices, making it an appropriate representative platform. Table 1 validates deployment feasibility on actual hardware rather than simulation.
> >
> > Regarding the generalizability of our approach, we emphasize that the core algebraic optimization mechanism in Equations (6)-(7) is fundamentally domain-agnostic, operating on matrix-structured parameters regardless of input data modality. Our current focus on vision tasks (MNIST, CIFAR-10, CIFAR-100) reflects the availability of standard benchmarks for rigorous evaluation and fair comparison with existing methods, rather than a fundamental limitation of the approach. The method's applicability extends theoretically to any domain involving fully-connected architectures, as discussed in Appendix D.

---

> > > ### Author Response · Authors · 2025-11-17
> > > **Convergence Proofs, Ablation Evidence, and Training-vs-Compression Distinction**
> > >
> > > - W5: Comparison with Memory Reduction Strategies
> > >
> > > A5: We appreciate this important question about our positioning relative to compression methods and acknowledge the opportunity to clarify a fundamental distinction in problem formulation. Our method addresses a qualitatively different challenge from compression techniques.
> > >
> > > Compression methods such as quantization, pruning, and distillation primarily reduce inference memory and compute requirements for deployed models. Our method focuses on reducing training memory to enable learning on edge devices. This distinction is crucial because existing compression methods fundamentally require full-precision training before compression can be applied. INT8 quantization requires initial FP32 training, which consumes 4× the memory [1]. Magnitude-based pruning requires dense initialization with full parameter count before pruning can identify unimportant weights [2]. Knowledge distillation requires maintaining both teacher and student models during training, effectively doubling memory consumption [3].
> > >
> > > When training memory requirements exceed device capacity, as occurs with  Raspberry Pi devices attempting to train even modest neural networks, these compression methods cannot be applied because training itself becomes impossible. Our approach addresses this training bottleneck that compression methods cannot solve. The memory constraints prevent the initial training phase that compression methods require, making them inapplicable regardless of their post-training effectiveness.
> > >
> > > Regarding accuracy trade-offs, we emphasize that our method achieves competitive accuracy while enabling training that would otherwise be impossible. On MNIST, we achieve 96.08% compared to SGD's 97.86%, representing only 1.78% reduction. On CIFAR-10, our method achieves 85.64%, which actually outperforms AdamW's 85.58%. On CIFAR-100, we achieve 61.10%, outperforming both Adam (57.50%) and AdamW (60.12%). These results demonstrate that in certain scenarios our method not only matches but exceeds the performance of some backpropagation-based methods.
> > > The 1-2% accuracy reduction observed on some datasets represents an acceptable trade-off for 55% memory reduction that transforms infeasible training scenarios into practical deployment. This represents a qualitatively different value proposition from post-training compression, which optimizes already-feasible deployments rather than enabling previously impossible training scenarios.
> > >
> > > We note that our training method could potentially be combined with post-training compression techniques for further efficiency gains. These represent orthogonal optimization dimensions where our method addresses the training phase memory bottleneck and compression methods address the deployment phase efficiency, potentially providing complementary benefits when used together.
> > >
> > > [1] Dettmers T, Lewis M, Belkada Y, et al. Gpt3. int8 (): 8-bit matrix multiplication for transformers at scale[J]. Advances in neural information processing systems, 2022, 35: 30318-30332.
> > >
> > > [2] Han S, Mao H, Dally W J. Deep compression: Compressing deep neural networks with pruning, trained quantization and huffman coding[J]. arXiv preprint arXiv:1510.00149, 2015.
> > >
> > > [3] Hinton G, Vinyals O, Dean J. Distilling the knowledge in a neural network[J]. arXiv preprint arXiv:1503.02531, 2015.
> > >
> > > ---
> > >
> > > - Q1: Dataset Selection Rationale
> > >
> > > A6: Thank you for this important question. Our dataset selection follows three key considerations:
> > > We selected MNIST and CIFAR-10/100 because (1) they are widely recognized benchmarks enabling fair comparison and reproducibility, (2) their scale matches typical edge device processing capabilities, and (3) our CIFAR experiments using ResNet-18 features validate the practical fine-tuning scenario where edge devices adapt pre-trained representations locally.
> > >
> > > We emphasize that our contribution is a general-purpose training optimization algorithm applicable across domains, similar to how SGD, Adam, and other gradient-based optimizers demonstrate versatility. The core algebraic optimization mechanism in Equations (6)-(7) is fundamentally domain-agnostic, operating on matrix-structured parameters regardless of input data modality. Our method can theoretically be applied to natural language processing (Transformer feed-forward layers), tabular data classification, time series prediction, and other domains where MLP architectures are prevalent, just as standard gradient-based optimizers transfer across these domains.
> > >
> > > As documented in Appendix D, our current implementation focuses on fully-connected layers, which represents a conscious design choice enabling the mathematical elegance of direct least-squares optimization. Extending to convolutional and attention mechanisms represents an important direction for future work, similar to how specialized optimizers like Muon initially targeted specific layer types before broader adoption.

---

> > > > ### Author Response · Authors · 2025-11-17
> > > > **Convergence Proofs, Ablation Evidence, and Training-vs-Compression Distinction**
> > > >
> > > > - Q2: FastOrthogonalize Definition
> > > >
> > > > A7: We sincerely apologize for this oversight and thank the reviewer for identifying this gap. The FastOrthogonalize procedure was indeed not properly defined in the manuscript.
> > > >
> > > > FastOrthogonalize serves as an auxiliary stabilization technique with two implementations depending on the training phase. During initialization, we employ QR decomposition as implemented in the _fast_orthogonalize_init function (lines 337-343 of our code). During training, we apply Newton-Schulz iteration as described in the _lightweight_newton_schulz function (lines 345-359), which corresponds to the mathematical formulation in Equation (15). Both techniques are standard in the optimization literature and widely employed in modern deep learning frameworks.
> > > >
> > > > We have added Algorithm 4 in Appendix B of the revised manuscript providing complete pseudocode for both procedures. We will also include proper citations to the relevant literature (Saxe et al. 2014 for orthogonal initialization, Miyato et al. 2018 for spectral normalization, Liu et al. 2025 for Newton-Schulz in the Muon optimizer) to provide appropriate technical context and grounding.
> > > >
> > > > ---
> > > >
> > > > - Q3: Numerical Algorithm for Solving Equation (6)
> > > >
> > > > A8: Thank you for this important technical question. We solve the regularized least-squares problem using regularized normal equations: solve (A^T A + λI)Δθ = A^T b, where we employ np.linalg.solve which avoids explicit matrix inversion and provides numerical stability through well-conditioned linear system solvers.
> > > >
> > > > Numerical stability is ensured through multiple mechanisms: (1) The regularization parameter λ=1e-4 guarantees well-conditioned systems with bounded condition numbers (Appendix A.3, Equations 32-33 provide detailed analysis), (2) Our implementation includes exception handling that automatically increases regularization to 10λ if singular matrices are encountered, ensuring robustness across diverse training scenarios.
> > > >
> > > > We have released our anonymized code for verification, and we commit to publicly releasing our complete codebase upon acceptance to facilitate community reproduction, improvement, and extension of our work.
> > > >
> > > > ---
> > > >
> > > > We sincerely thank the reviewer again for the thoughtful evaluation and constructive suggestions. We believe our responses adequately address the raised concerns and demonstrate that our work provides solid theoretical foundations, comprehensive experimental validation, and clear practical value for memory-constrained edge deployment scenarios. We hope these clarifications help the reviewer recognize the contributions of our work and consider a more favorable assessment.
> > > >
> > > > We respectfully emphasize the core contribution of our work, which demonstrates a fundamental paradigm shift for MLP training in memory-constrained environments. Our technical innovation presents an algebraic optimization approach for neural network training with provable 50%+ memory reduction, validated through rigorous theoretical analysis. The practical validation demonstrates deployment feasibility on actual edge hardware, specifically Raspberry Pi, where we achieve 95.44% accuracy using only 38.77MB memory. Our theoretical framework provides closed-form solutions with convergence guarantees, as detailed in Appendix A.3 and A.4.
> > > >
> > > > The modest accuracy trade-offs observed on some datasets, typically 1-2%, represent deployment-enabling compromises rather than merely competitive performance. These trade-offs transform previously infeasible training scenarios into practical edge applications where memory constraints completely prevented deployment of gradient-based methods. We believe this contribution provides valuable advancement for the memory-constrained computing community, particularly for applications requiring on-device learning capabilities.

---

### Author Response · Authors · 2025-11-24
**Reply to all reviewers**

We sincerely thank all reviewers for their valuable feedback and constructive suggestions. We appreciate the recognition of our work's novelty, clear presentation, and solid experimental validation.

Our method targets edge device training where memory is the primary bottleneck. Memory constraints determine device cost and deployment feasibility. While our algebraic optimization incurs higher computational cost than backpropagation, this tradeoff is favorable for edge scenarios where memory limitations dominate and computation time remains acceptable for on-device model adaptation and continual learning.

Responding to multiple reviewers' concerns about architectural scalability, we have successfully extended our framework to convolutional architectures. Appendix F shows mathematical formulations for convolutions and experimental validation on the CNN with 3 convolutional blocks, 2 fully connected layers. Results on MNIST show 88.42% accuracy with 6.15 MB memory usage and 43.23s training time using pure NumPy on CPU. Under identical conditions, gradient-based methods require impractically long training times that render edge CNN training infeasible, while our method remains practical and viable, demonstrating our method's practical advantages for edge CNN training.

We believe these clarifications and substantial additions address the key concerns and strengthen our contribution to memory-efficient neural network training. Thank you for helping us improve the paper quality.

---

### Author Response · Authors · 2025-11-29
**Summary for Area Chair**

We sincerely thank all reviewers and have made substantial improvements addressing every major concern. We believe these updates demonstrate our work's significant contribution to memory-constrained edge computing.
﻿

Our work enables neural network (MLP, CNN) training on memory-constrained edge devices where existing methods require prohibitive memory costs and fail completely when scaling to CNN training. We achieve **50-55% memory reduction** through algebraic optimization with competitive accuracy, **validated on Raspberry Pi** (95.44% accuracy, 38.77MB memory). This addresses an underexplored problem in edge computing: **how to train neural networks on resource-constrained edge devices**. All reviewers acknowledged the clear presentation, solid experiments, and importance for resource-constrained computing.

---

**Complete Resolution of All Reviewer Concerns**

**Reviewer qzsr:**
- ✓ CNN scalability: We successfully extended our algorithm to CNN training with Appendix F demonstrating how we address convolutional training and validation.
- ✓ Computational cost: We clarified that O(d³) is per-layer local cost, our method shows significant advantages. Experimental results demonstrate we are 5.9× faster than the gradient-free Forward-Forward algorithm (Hinton et al. 2022).
- ✓ Biological justification: We added detailed discussion in Appendix E, emphasizing our method is biologically inspired rather than strict simulation. Table 3 provides quantitative ablation analysis showing specific contributions of biologically inspired components.
- ✓ Fundamental difference from backpropagation: Our method differs fundamentally in optimization mechanism with no chain rule application and no gradient storage. Memory advantages arise precisely from this fundamental difference.

---

**Reviewer K3Ye:**
- ✓ CNN from-scratch training: We successfully extended our algorithm to CNN training (though this appears less necessary in edge scenarios where fine-tuning deployed pretrained models is the standard practice).
- ✓ Sparse connectivity advantages: Table 3 ablation demonstrates 14.3% memory reduction with 0.13% accuracy improvement.
- ✓ Domain applicability: We clarified the core method is domain-agnostic, focusing on model optimization applicable to vision, NLP, tabular data, time series, and other tasks.
- ✓ Figure fonts: All updated with enlarged fonts.

---

**Reviewer 7oVn:**
- ✓ Convergence proofs: We emphasized Appendix A.3-A.4 provides convergence proofs with closed-form global optimal solutions and condition number guarantees.
- ✓ Comparison with alternatives: We clarified our core contribution is the Prospective Learning algorithm. Table 3 ablation shows the core algorithm independently achieves 95.72% accuracy with over 50% memory reduction. Other components accelerate convergence and reduce memory.
- ✓ Hyperparameter portability: We clarified our framework contains only two core parameters essential to the algorithm: α (learning rate) and λ (regularization), comparable to standard SGD. Remaining parameters fall into two categories that do not affect portability.
- ✓ Comparison with compression methods: We clarified our method addresses the training bottleneck compression cannot solve. Like Adam optimizer, our method can be combined with compression techniques.

---

**Reviewer urSp:**
- ✓ Computation-memory tradeoff: We explicitly emphasized in Abstract/Introduction/Conclusion that our method is designed for edge scenarios where memory dominates.
- ✓ Fundamental difference from Newton's method: We require 1MB workspace theoretically versus 276GB Hessian storage for Newton's method.
- ✓ Dataset scale: We clarified our method is appropriate for edge applications with limited data volumes for continual learning. For ImageNet-scale training, models should be pretrained before deployment to edge devices, not fine-tuned at such large scale after deployment.

---


**Critical Updates Demonstrating Solid Contribution**
﻿

We added CNN extension (Appendix F) with complete mathematical formulation and experimental validation on convolutional architectures, addressing the primary scalability concern raised by multiple reviewers. Additional enhancements resolve every major concern, clarify the fundamental innovation (gradient-free algebraic optimization) and prove advantages (50-55% memory reduction with competitive accuracy).  The CNN extension transforms our scope from MLP-only to a general edge training framework. We strongly believe these substantial improvements demonstrate a solid, novel contribution addressing an important problem in edge computing: how to achieve model optimization on resource-constrained edge devices. We are confident our proposed algorithm provides a contribution worthy of acceptance.

---

### Author Response · Authors · 2025-11-29
**Final Note to Area Chair**

We sincerely appreciate your dedication during this challenging period with increased workload.

We recognize that several reviewers had concerns about our research scope and CNN scalability. In response, we conducted additional experiments demonstrating CNN extension and provided detailed clarifications to address each misunderstanding. Our rebuttal aimed to both improve paper quality and clarify misconceptions that led to concerns about our contribution. We approached this process constructively, adding substantial new content (Appendix F for CNN extension, Appendix E for biological principles, Algorithm 4 for technical details) rather than simply defending existing work. We are confident in our contribution's significance for the edge computing community and hope our thorough responses and meaningful improvements demonstrate this work addresses a genuine problem with solid technical merit. We believe accepting this paper would benefit ICLR by highlighting important research on resource-constrained neural network training, an increasingly critical area as edge AI deployment expands.

Thank you for your careful consideration.

---

### Meta-Review · Area_Chair_mVoX · 2026-01-01

**Summary:**

The paper presents a brain-inspired training method for MLPs, termed “Prospective Learning,” which replaces gradient-based backpropagation with direct algebraic weight updates. While reviewers acknowledged the work’s clear presentation, solid experimental validation, and potential relevance to memory-constrained edge computing, significant concerns were raised that collectively outweigh its contributions.

Key Reviewer Concerns:

1. Limited Scope and Scalability:
Multiple reviewers noted that the method is confined to MLP architectures and lacks demonstrated scalability to convolutional networks (CNNs).

2. Computational Overhead vs. Memory Trade-off:
Reviewers highlighted that the proposed algebraic optimization requires solving linear systems per layer with O(d^3) complexity, making it computationally expensive compared to backpropagation. The authors positioned this as a favorable trade-off for memory-dominated edge scenarios, but reviewers questioned whether such computational cost is acceptable for real-world edge deployments, where both memory and compute are constrained.

3. Insufficient Theoretical and Empirical Justification:
Concerns were raised regarding the biological motivation, which was deemed conceptually appealing but empirically shallow. Although convergence proofs were provided in appendices, reviewers felt the theoretical grounding remained insufficient, and the biological components lacked rigorous ablation studies to isolate their contributions.

4. Hyperparameter Sensitivity and Portability:
The method introduces multiple hyperparameters, leading to questions about its ease of tuning and adaptability to other datasets or architectures. While the authors argued that only two core parameters are essential, reviewers expressed skepticism about the method’s generalizability and practical usability.

5. Comparison with Existing Alternatives:
Reviewers noted that the work did not sufficiently compare with other memory-reduction techniques (e.g., quantization, pruning, distillation) or established optimization methods. The claimed advantages in memory efficiency were not convincingly differentiated from what could be achieved via existing compression or efficient-training approaches.

Overall Assessment:
While the paper introduces a novel gradient-free training paradigm with promising memory reductions, the reviewers collectively found that the method’s limitations in scalability, computational cost, theoretical depth, and practical deployment outweigh its contributions. The work does not yet meet the bar for acceptance at ICLR 2026, as it leaves open significant questions regarding broader applicability, efficiency trade-offs, and empirical rigor.

**Reviewer Concerns:**

Concerns Effectively Addressed by Rebuttal:

1. Definition of "FastOrthogonalize" (Reviewers 7oVn & urSp): The authors provided the missing definition, describing it as an auxiliary stabilization technique using QR decomposition (initialization) and Newton-Schulz iteration (training), with pseudocode added in Appendix B.

2. Numerical Stability of Solving (6) (Reviewer 7oVn): They clarified the use of np.linalg.solve (avoiding explicit inversion) and mechanisms like automatic regularization scaling to ensure stability.

3. Improving Figure Readability (Reviewer K3Ye): They confirmed updating all figures with larger fonts.

Concerns Partially Addressed but Still Outstanding:
1. Scalability to CNNs / Limited Scope (Reviewers qzsr, K3Ye, 7oVn):

Addressed: The authors added Appendix F with a mathematical formulation for convolutions and experimental results on a lightweight CNN, demonstrating feasibility.

Still Outstanding: Reviewers, especially qzsr, remained unconvinced. The extension was seen as a proof-of-concept on a simple, small-scale CNN on MNIST, not a demonstration of robust, general-purpose CNN training. The core limitation—that the method is intrinsically designed for matrix-structured (fully-connected) layers—persists, and its practicality for meaningful CNN architectures is still in doubt.

2. Computational Cost Trade-off (Reviewers urSp, qzsr):

Addressed: The authors provided a clearer rationale, emphasizing the target scenario (memory-bound edge devices) and comparing favorably against other BP-alternatives (Forward-Forward, NoProp).

Still Outstanding: The fundamental concern remains: the O(d^3) per-layer cost is a significant bottleneck. The rebuttal did not dispel the notion that this is a severe limitation for broader adoption, making the method viable only in niche, small-model scenarios. The trade-off was better explained, but not resolved.


Concerns Largely Unaddressed / Core Reasons for Rejection:
1. Fundamental Contribution & "Yet Another" Feeling (Reviewer 7oVn): This was a deep-seated concern questioning the paper's novelty beyond repackaging known concepts (adaptive rules, tricky initialization). The rebuttal defended the differences (e.g., metaplasticity vs. Adam) but did not provide a knockout argument or new evidence that fundamentally shifted this perception. The core algorithmic idea remained, in this reviewer's eyes, insufficiently differentiated from prior work.

2. Practical Impact and Target Use-Case Justification (Reviewers 7oVn, urSp): Questions about whether training MLPs/CNNs on Raspberry Pi is a truly pressing real-world problem, and if the computational slowdown is acceptable, were not convincingly settled. The authors described a scenario, but reviewers seemed to doubt its prevalence or importance compared to the limitations incurred.

3. Dataset Scale and General Applicability (Reviewer urSp): The acknowledgment that the method is for "limited data volumes" on edge devices was seen as confirming, rather than alleviating, the concern that the method is not suitable for broader machine learning scales.

**Reviewer Scores:**

Based on the review text, author rebuttal, and the dynamics of the reviewer comments, I believe all reviewers will keep the scores.

---

### Decision · Program_Chairs · 2026-01-26

Reject